



# Numerical Study of the Unsteady Blade Root Aerodynamics of a 2MW Wind Turbine Equipped With Vortex Generators

Ferdinand Seel[1], Thorsten Lutz[1], and Ewald Krämer[1]

[1]University of Stuttgart, Institute of Aerodynamics and Gas Dynamics (IAG), Pfaffenwaldring 21, 70569 Stuttgart, Germany

**Correspondence:** Ferdinand Seel (ferdinand.seel@iag.uni-stuttgart.de)

**Abstract.** In order to design vortex generators for modern multi-megawatt wind turbines accurately, the three-dimensional behaviour of the boundary layer has to be considered. Due to the rotation of the blade, the lift-enhancing rotational augmentation has a considerable impact, especially in the inner blade sections. To investigate the interaction of vortex generators and rotational augmentation, high-fidelity computational flow simulations by means of unsteady Reynolds-averaged Navier-Stokes methods are presented for a rotating blade of a generic 2 MW horizontal axis wind turbine. The inner blade section is analysed with and without vortex generators for two different pitch settings including one causing largely separated flow. Two ways of placement of the vortex generators on the blade with different radial starting positions are investigated in order to find out if a coexistence of the two lift-enhancement methods (i.e. rotational augmentation and vortex generators) is beneficial. All simulations are performed with the flow solver FLOWer and the vortex generators are modelled by the introduction of source terms into the computational domain through a BAY-type model. For the case without vortex generators, it is found that the strength of rotational augmentation largely depends on the effective angles of attack (i.e. the pitch setting). For the case with lower effective angles of attack, rotational augmentation is a cyclic phenomenon whereas for the case with higher effective angles of attack it generates large loads in the inner root section due to a constant centrifugal pumping mechanism in time. The results from the cases with vortex generators display a rather destructive interaction of vortex generators and rotational augmentation on the torque. For low effective angles of attack and thus attached flow conditions, vortex generators exhibit slight losses compared to the case without VGs as they inhibit spanwise flow through rotational augmentation. For high effective angles of attack, the vortex generators placed over 30% of the blade produce an increase of 3.28% in torque compared to the case without VGs and high centrifugal pumping.

## 1 Introduction

The trend in the development of blades for new horizontal axis wind turbines (HAWT) is towards ever greater blade lengths, which leads to new structural requirements in the design of the blade root region. One of these is the requirement of high relative airfoil thicknesses. Their disadvantages in terms of flow separation can be reduced by vortex generators (VGs) and is therefore expected to become an integral part of the blade design in future.

Those vane-like passive devices, placed on the blade surface at a certain inclination angle $\beta$ to the local inflow, create streamwise vortices which reenergize the boundary layer by transporting high momentum flow from the outer boundary layer



towards the inner boundary layer. This mechanism delays separation and enables the boundary layer to overcome stronger adverse pressure gradients. Thus, VGs permit to reach higher maximum lift coefficients at higher angles of attack. Nowadays, VGs are optimized through wind tunnel tests and numerical simulations both on extruded airfoil sections to perform ideally in combination with the specific radial airfoil sections. Through an iterative process, the numerous geometrical parameters like VG height, inter- and intraspacings between the VGs and chordwise positions on the airfoil section are optimized. The influence of VGs on the boundary layer in a 2D and non-rotating flow has already been studied since the first application for diffuser flows in the 1940s (Taylor et al., 1947) for many different fields and for different applications, e.g. delay of flow separation (sub- and supersonic), increase of heat transfer or reduction of noise. Even for wind energy application (i.e. airfoils with high relative thicknesses), the effect of VGs on the aerodynamics of airfoil sections has been studied in detail as part of the extensive efforts carried out, for example, by the EU project AVATAR (Baldacchino et al., 2018, 2016).

On the rotor blade, VGs act differently, due to the rotating motion and the consequently upcoming centrifugal force inducing spanwise flow. Only a few authors focused on an aerodynamic investigation of the effect of VGs on rotor blades by means of Computational Fluid Dynamics (CFD) methods. Troldborg et al. (2016) investigated the DTU 10 MW wind turbine equipped with BAY-modelled VGs on the inner part of the blade. They observed that the performance of blade sections equipped with VGs was affected by the presence of VGs on other sections. Furthermore, it was qualitatively shown that VGs reduce the lift enhancement due to rotational effects at high angles of attack. Zhu et al. (2021) investigated the stall-regulated NREL Phase VI blade with fully-resolved VGs but chose an operational condition with leading edge separation. In this case, the VGs are within the separated flow domain and were found to reduce the rotational effects and aggravating the size of the separated domain. Recently, Manolesos et al. (2023) studied VGs on a tidal rotor both experimentally and numerically with BAY-modelled VGs. As HAWT and tidal turbines have similarities in operating conditions, they selected a VG setup based on HAWT best-practices and showed the need to include the rotating motion into the VG design processes. By comparing controlled (i.e. equipped with VGs) and uncontrolled (i.e. without VGs) cases for the turbine, it was shown that, particularly for high rotational speed and low Reynolds numbers (model scale), the rotational forces are stronger (Gross et al., 2012) and thus already reduce separation making the VGs less useful than expected from results of the airfoil without rotation. Baldacchino (2019) mentioned that the imbalance occurring between VG vortices of an array due to skewed inflow, as it occurs in blade root sections, can affect the performance of the VGs. These results emphasize the need of numerical methods to obtain an optimal VG design for the rotor blade.

## 1.1 Rotational Augmentation

The positive effect of rotation on the boundary layer (i.e. rotational augmentation) was first discussed by Himmelskamp (1945) who observed an increase in lift and a delay of separation on a propeller blade compared to the same airfoil in non-rotational 2D flow. This so-called "Himmelskamp effect" was explained by the rotational forces acting on the boundary layer of the blade. In Fig. 1 the different mechanisms of the effect are shown: The centrifugal force, pointing towards the blade tip and increasing with the distance from the center of rotation, accelerates the flow in spanwise direction (frame 2 in Fig. 1) and increases the momentum inside the boundary layer. As the centrifugal force increases towards the blade tip, each blade section loses more



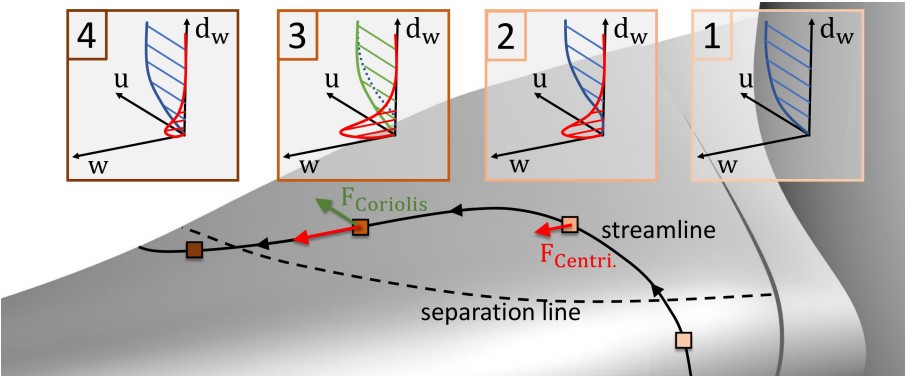

**Figure 1.** Schematic description of rotational augmentation.

mass flow towards the tip than it gets from the inner sections (Himmelskamp, 1945). Thus, a suction effect called centrifugal pumping towards the blade tip is installed. On top of this effect, the radial velocity component in the rotating coordinate system creates a Coriolis force which points towards the blade trailing edge. This force reduces the extent of the separated areas and reenergizes the chordwise boundary layer (frame 3 in Fig. 1) to withstand higher adverse pressure gradients (Bangga, 2018). The rotational forces acting on the nearly separated or separated flow also highly impact the vortex shedding frequencies on the blade. The results from Schreck (2010) show that the vortex shedding frequency is largely reduced, compared to the usually expected Strouhal numbers of around 0.15 to 0.2 for separated airfoils in non-rotating conditions. Depending on the operational regime of the turbine, this leads to a different unsteady aerodynamic loading of the blade and fatigue loads. Those findings emphasize the fundamental differences between the flow dynamics on the rotating blade and extruded airfoils.

Herraéz et al. (2016) stated that the main driver for centrifugal pumping is the centrifugal force acting on the boundary layer. The radial pressure gradient has only a low impact on the radial flow (Bangga, 2018). Nevertheless, it is important to mention that rotational effects heavily depend on the turbine (Herráez et al., 2014). Thereby, according to Bangga (2018), a large dependency is attributed to the chord to radius distribution $c/r$ of the blade: The influence of rotational augmentation alleviates with decreasing $c/r$ until it vanishes ($c/r < 0.1$). For this reason, the major rotational effects in this work will first be analysed for the uncontrolled blade before being studied in combination with vortex generators which also take influence on the boundary layer flow.

Regarding the quantification of rotational augmentation, it is common to compare inner blade sections with the corresponding 2D sections (extruded airfoils) (Bangga, 2018; Herráez et al., 2014). This requires an extraction of the relevant inflow parameters like inflow velocity, Reynolds number and effective angle of attack from the 3D case as input for the 2D cases. The estimation of the effective angle of attack is not trivial. It can be determined by means of an algebraic equation (Herráez et al., 2014) or by using the 3D flow field (Bangga, 2018) and one of several extraction methods proposed and evaluated by Jost et al. (2018). Unfortunately, all those methods are subjected to large uncertainty particularly for the inner part of the blade due to the large separated areas. On top of the inaccuracies from the determination of the angle of attack, the 2D vs. 3D comparison



includes further inaccuracies as largely separated flow is a complex 3D phenomenon where methods based on the integral boundary layer equation or steady Reynolds-averaged Navier Stokes (RANS) CFD simulations are prone to fail. Computations

of higher fidelity methods like Detached Eddy Simulation (DES) are expensive and thus not a feasible alternative.

In this work, a method proposed by Bangga (2018) will be used. It allows to quantify the Himmelskamp effect without using additional 2D methods but by defining the Himmelskamp force as

$$F_H = F_{Coriolis} + F_{centri} = -2 \cdot (\boldsymbol{\Omega} \times \boldsymbol{V}) + \boldsymbol{\Omega} \times (\boldsymbol{\Omega} \times \boldsymbol{r}) \tag{1}$$

with $\boldsymbol{\Omega}$ being the rotational speed, $\boldsymbol{r}$ the distance of the regarded point to the center of rotation and $\boldsymbol{V}$ the velocity of the fluid

in the rotating frame of reference.

## 1.2 Vortex generators

VGs are widely used in wind energy due to their robustness, low production costs and installation costs. They become even more interesting because various studies provide guidelines for numerous geometrical parameters i.e. VG shape, height, length, inclination angle, spacings in spanwise and chordwise position.

Regarding the VG shape, it is established to use triangular shaped VGs as they offer a good compromise between the drag penalty and the strength of the shed main vortex (Godard and Stanislas, 2006). As the large drag penalty for rectangular VGs mainly results from the leading edge separation on the VG suction side (Hansen et al., 2016), the so-called cropped-delta shapes also show a good performance: The cropping leading edge reduces the drag and in comparison to a triangular VG the vortex strength increases due to the larger surface area (Seel et al., 2022).

VGs are only efficient when acting inside the boundary layer. Hence, their height should not exceed the boundary layer height $\delta_{99}$. Consequently, VGs are very small devices and thus the influence area of the shed vortex is limited to a small spanwise extent. In order to obtain the positive mixing effect over a large span, a large number of VGs has to be placed along the blade. Regarding the relative position of the VGs towards their neighbors, Baldacchino et al. (2018) showed that VGs placed in counter-rotating pairs, directing the flow in between the pair towards the airfoil surface (i.e. common-down flow),

are the most efficient. To achieve such a setup, the interspacing (spacing between the VG pairs) is chosen to be much larger than the intraspacing (spacing between two VGs of a pair). However, the angle of the incoming flow is very relevant for this VG placement: As shown by Baldacchino (2019), who investigated experimentally counter-rotating common-down flow VG arrays placed in an skewed angle to the local flow, the vortex dynamics and the vortex breakdown is different compared to the case without skew angle. The asymmetric strength and trajectory of the shed vortices form strong and weak vortex pairs, and

the unbalanced vortex strengths finally lead to a streamwise motion of the entire vortex array as it is known from co-rotating VG arrays. This skewed inflow conditions occur in the blade root area as the flow velocity has a component pointing towards the blade tip.

In light of the above, it is clear that when it comes to CFD modelling of VGs on a rotor blade, large meshes and high computational resources are required as multiple very small scale flow mechanisms (vortex system shed by each VG) influence

much larger flow mechanisms (flow separation on the rotor blade). For this reason, the refinements of the mesh have to be





chosen with care. Seel et al. (2021) showed that CFD modelling of fully-resolved VGs and RANS methods requires highly resolved blade meshes not only in the VG area but also along the propagation path of the main VG vortices. In addition, each VG needs to be fully-resolved leading to a large meshing workload and a further raise in number of cells. To reduce these two drawbacks, a method based on the so-called BAY model is utilised in this work. It uses source-terms applied into the

computational domain deflecting the flow to mimic the impact of the VGs. The model was initially proposed by Bender et al. (1999), and a variant of it proposed by Seel et al. (2022) is used in this work and explained in more detail in Sect. 2.2. Among the main VG vortex, which is the main driver of the momentum transfer in the boundary layer, several other smaller vortices form in its vicinity. Their emergence depends on the VG height $h_{VG}$ and on the vane inclination angle $\beta$ (Velte et al., 2016). These smaller vortices influence the main vortex and have an impact on the efficiency and durability of the momentum transfer

of the main VG vortex. Therefore, Seel et al. (2022) showed that the mesh resolution can be reduced for the proposed BAY model derivative but a large refinement of the cell size of about $1/8$ to $1/16$ of the VG height in the VG area and propagation area is recommended in span- and chordwise direction in order to achieve a good agreement with the vortex system of the fully-resolved case.

### 1.3 Scope of work and objectives

In the first part of the introduction, the physics of two ways of increasing the lift coefficient in the inner part of the blade were presented. On the one hand, the positive effect of the so-called rotational augmentation (i.e. centrifugal pumping and Coriolis force) was extensively investigated by the community (Himmelskamp, 1945; Schreck, 2010; Gross et al., 2012; Herráez et al., 2014; Herraéz et al., 2016; Bangga, 2018) and the positive influence on the lift over a large range of angles of attack (from fully attached to detached flow) was shown. On the other hand, VGs are able to delay the onset of boundary layer separation

and thus increase the lift of extruded wind turbine airfoils for high angles of attack which was clearly shown in a multitude of experimental and numerical studies (Velte and Hansen, 2013; Manolesos and Voutsinas, 2015; Baldacchino et al., 2016, 2018). Hence, both ways alleviate the separated flow in the inner part of the blade. However, the first (i.e. rotational augmentation) is clearly advantageous in the uncontrolled 3D flow with rotation and the second (i.e. VGs) has shown its benefit in 2D flow. Eventually, both ways are applied on a 3D blade which encounters all the corresponding and already introduced effects.

The objectives of this work, are formulated by the following questions:

Q1: To what extent are the loads and the state of the boundary layer of the considered turbine affected by rotational augmentation for the uncontrolled case (without VGs) for different pitch settings of the blade?

Q2: How do the streamwise convecting VG vortices interact with the radial flow caused by rotational augmentation? In this context, it will be shown whether the interaction is constructive or destructive and if a combination of both effects is

expedient.

Q3: How and for which considered cases does rotational augmentation trigger flow unsteadiness?





In order to to answer the questions posed, one uncontrolled setup and two different VG setups, starting at different radial positions along the blade root are investigated. In Sect. 3, the uncontrolled case is evaluated to understand the rotational augmentation effects which occur on this particular blade in the uncontrolled state. In Sect. 4, the interaction of rotational augmentation and VG vortices further outside in radial direction is investigated by leaving the inner root section uncontrolled. Finally, in Sect. 5, the blade root is studied in a fully controlled setup.

## 2 Numerical set-up

The numerical CFD setup is based on a generic pitch-controlled $2\,\mathrm{MW}$ HAWT called I82 introduced by Arnold et al. (2020) with a rotor radius of $R = 41\,\mathrm{m}$. In order to reduce the high computational costs related to the consideration of VGs, a so-called one-third model is used with $120°$-periodic boundary conditions. Therefore, only one blade, one-third of the hub and of the nacelle are meshed but in return a very high mesh resolution is possible in the area of interest (i.e. inner sections of the blade suction side). The influence of the simplifications from the one-third model (i.e. no blade tower interaction, no atmospheric boundary layer and no tilt angle) on the studied phenomena in this work is considered negligible.

### 2.1 CFD solver: FLOWer

The flow simulations are performed with the block-structured CFD solver FLOWer, which was initially developed by the German Airspace Center (DLR) (Kroll et al., 2002). The code solves the three dimensional compressible RANS equations in integral form. In this work, the time accurate dual time-stepping technique is employed. For turbulence modelling, the Menter-SST two-equation eddy viscosity model (Menter, 1994) is used and surfaces are considered fully-turbulent. The implemented chimera overlapping mesh technique, which allows to combine separate meshes, is very important for the present work as the extent of the considered length scales is large. The smallest considered scales of the flow field start from the vortex system shed by the VGs towards the largest scales including the induction and wake of the blade.

### 2.2 BAY Model

In order to study the overall impact of a high number of VGs on a wind turbine blade ($> 100$) with reasonable computational costs, a BAY-type model was implemented in FLOWer and compared to fully-resolved VG simulations and validated with experimental data (Seel et al., 2022). The implementation is very similar to the so-called actuator shape BAY model proposed by Réthoré et al. (2014) and used for VGs by Troldborg et al. (2016) which aims to improve the representation of the VG shape compared to the classical BAY model (Bender et al., 1999) or the jBAY model (Jirásek, 2005). The implementation in FLOWer is also able to consider any kind of 2D VG shape, but the introduction of forces is not done via an actuator shape model (Réthoré et al., 2014) but via kernel points as part of a point cloud representing the VG shape. Starting from those kernel points, the force is smeared into the computational domain in order to guaranty a high level of numerical stability and reduce the grid dependency. The BAY force $f_{i,j}$ for each kernel point $p_{i,j}$ is computed as

$$\overrightarrow{f_{i,j}} = C_{VG} \cdot \rho \cdot |\boldsymbol{u}|^2 \cdot S_{i,j} \cdot (\hat{\boldsymbol{u}} \cdot \boldsymbol{n}) \cdot (\hat{\boldsymbol{u}} \cdot \boldsymbol{t}) \cdot (\hat{\boldsymbol{u}} \times \boldsymbol{b})$$



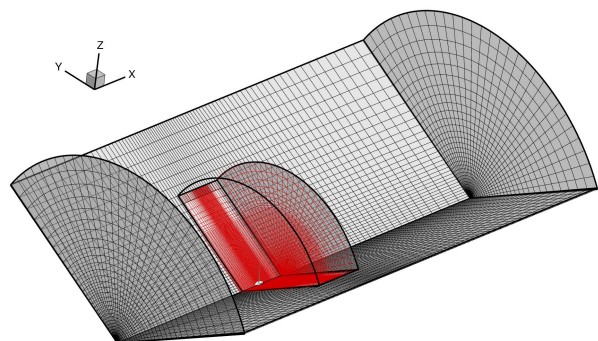

**Figure 2.** Computational background meshes of the one-third model setup with the refined area (red) to include the wind turbine. For better visibility, the number of cells was reduced by a factor of four.

with

$$\hat{\boldsymbol{u}} = \frac{\boldsymbol{u}}{|\boldsymbol{u}|} \tag{2}$$

where $C_{VG}$ corresponds to the BAY calibration constant (set to $C_{VG} = 10$ in this work according to Jirásek (2005)), $\rho$ corresponds to the density, $\boldsymbol{u}$ is the velocity vector and the vectors $\boldsymbol{b}$, $\boldsymbol{n}$ and $\boldsymbol{t}$ correspond to the spanwise, normal and tangential direction of the VG surface respectively. The redistribution of the forces is done via a Gaussian kernel distribution which leads to the force $F_c$ applied to each cell $c$ in the near field of a point $p_{i,j}$ of the CFD grid as

$$\boldsymbol{F_c} = \sum_{i=1}^{M} \sum_{j=1}^{N} \boldsymbol{f_{i,j}} \cdot \eta_{\varepsilon_{i,j}}^{3D}. \tag{3}$$

The smearing of the body forces is applied via the Gaussian kernels $\eta_{\varepsilon_{i,j}^{3D}}$ for each point. The implemented BAY model uses isotropic 3D Gaussian kernels defined as

$$\eta_{\varepsilon_{i,j}}^{3D}(d) = \frac{1}{\varepsilon_{i,j}^3 \pi^{\frac{3}{2}}} \exp\left[-\left(\frac{d}{\varepsilon_{i,j}}\right)^2\right]. \tag{4}$$

The parameter $\varepsilon_{i,j}$ corresponds to the Gaussian width and determines the spread and smearing of the force distribution. It is defined depending on the distance from the point $p_{i,j}$ towards the neighboring points and explained in detail by Seel et al.
(2022). Finally, the distance $d$ is defined between the considered cell of the CFD grid and the point $p_{i,j}$.

## 2.3 Meshes

The setup is composed of a total of nine meshes brought together with the chimera technique and contains a total of 114.2 million cells. As shown in Fig. 2, the 120°-background mesh (grey, 4.6 million cells) contains a refined background mesh (red, 9.6 million cells) to increase the resolution in the vicinity of the blade. The farfield radial distance as well as the upstream and
downstream distances were selected according to the best-practices for one-third model setups proposed by Sayed et al. (2015).





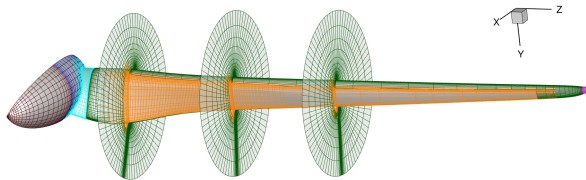

**Figure 3.** Computational meshes of the one-third model setup. For better visibility, number of cells is reduced by a factor of eight for the VG refinement mesh (orange) and by a factor of four for all other meshes.

In Fig. 3 the meshes of the wind turbine are shown. As the nacelle (brown, 0.37 million cells) is a non-rotating component, a specific boundary condition which subtracts the velocities resulting from the rotating motion in the first interior cell is used. The connector (light blue, 1.1 million cells) is linking the hub (dark blue, 0.37 million cells) and the blade (green, 11.3 million cells) and allows rotation of the blade around the z-axis to set the blade pitch. The blade mesh, equipped with an upwind facing winglet (magenta, 2.27 million cells) and embedded into a blade tip refinement mesh (not shown, 0.64 million cells) was provided by Wenz et al. (2022) and slightly adapted. The blade and winglet meshes have an O-topology and fulfill the requirements for RANS simulations with resolved viscous sublayer (i.e. $y^+ < 1$). Inside the boundary layer, a growth rate of 1.09 in wall-normal direction is used. Furthermore, a low conservative numerical error for thrust and torque was obtained during a convergence study made by Wenz et al. (2022). In order to model the effect of the VGs on the blade, the mesh resolution has to be very high in the VG area (minimum 8 cells per VG vortex in blade spanwise direction (Troldborg et al., 2015)). But also in the propagation area of the streamwise VG vortices, the resolution has to be resolved with cells of the size of at least $1/8$ of $h_{VG}$ (Seel et al., 2022) for the BAY model. For this purpose, a refinement mesh (Fig. 3 in orange) for the suction side of the blade was created. The mesh has 83.95 million cells in total and thus represents by far the largest mesh of the setup. In chordwise direction, it extends from 10% of the relative chord length to the trailing edge of the blade with 317 points. The point density is thereby increased at the position of the VG array and at the trailing edge. In radial direction, 2333 points are used from the innermost part of the blade to 90% relative blade length. In the area of the VG array, the spacing is kept at $\Delta z = 5\,\mathrm{mm}$ (i.e. $\Delta z = 0.000122 \cdot R$). At the end of the VG zone, towards the blade tip, the cell size in radial direction is increased smoothly over a relative distance of $0.1 \cdot R$. For the entire mesh, $y^+ < 1$ and the wall-normal growth rate in the boundary layer of 1.09 are identical to the values used for the blade mesh.

## 2.4 VG set-up description

As shown in Fig. 4, parabolically-shaped counter-rotating common-down flow VGs are used. The different geometrical parameters of the VGs, selected according to the best practices in literature (Baldacchino et al., 2018; Godard and Stanislas, 2006; Pearcy, 1961), are listed in Table 1. At this point it is mentioned that the parabolic shape is inspired by the results of an industry-led optimization of the VG shape not further addressed in this work. As VGs reach their highest efficiency when placed inside the boundary layer (i.e. $h_{VG} < \delta_{99}$) and because $\delta_{99}$ decreases along the radial blade direction towards the tip

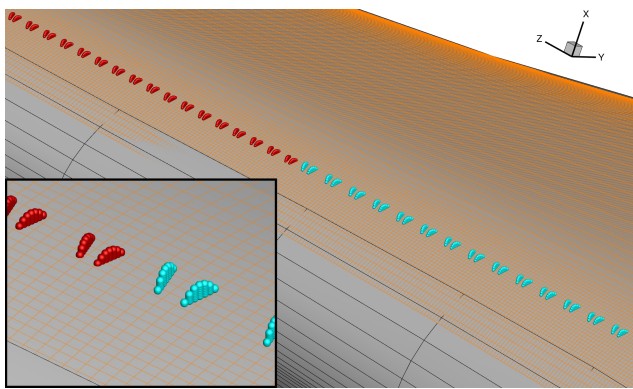

**Figure 4.** Placement of the VGs into the blade refinement mesh in orange. Kernel points of VGs belonging to array H16 with $h_{VG} = 16\,\mathrm{mm}$ (light blue) and array H11 with $h_{VG} = 11\,\mathrm{mm}$ (red). For better visibility, the number of cells was reduced by a factor of four.

due to the higher Reynolds numbers outboard, the VG height has to be adapted. In this work, two operational conditions with a different blade pitch angle (see Table 2), denoted in the following design (Des) case and off-design (oDes) case are considered. As the local $\delta_{99}$ distribution along the radial direction $z_{rel} = z/R$ (with $R$ the blade length) is different for each case, and to get an optimal $h_{VG}$-distribution along the blade for both operational conditions, $\delta_{99}(z_{rel})$ was extracted form the uncontrolled
CFD calculations for a chordwise relative region of $x_{VG}/c \pm 5\%$ and averaged for both cases and two revolutions (i.e. 1440 physical time steps). The resulting $\delta_{99}(z_{rel})$ distribution, as well as the chosen variation of $h_{VG}$ along the blade length, are plotted in Fig. 5. The variation of $h_{VG}$ along the span was adapted as a best fit of the finding from Godard and Stanislas (2006) who proposed $h_{VG} \leq 0.5\delta_{99}$ and Baldacchino et al. (2018) who proposed a dependency based on the chord length $c$ as $h_{VG} < 0.1c$. A step adaptation of the VG height is common instead of a linear one because it is closer to a realistic application
where it is usual to have only a few discrete VG heights available through mass-produced parts.

**Table 1.** Geometrical parameters of the VGs

| Configuration | Ctrrot. CD flow |
| --- | --- |
| height $h_{VG}$ [mm] | 22; 16; 11 $\approx \delta_{99}$ |
| Vane angle $\beta$ [°] | 15 |
| Length $l/h_{VG}$ | 3 |
| Intraspacing $d/h_{VG}$ | 3.8 |
| Interspacing $D/h_{VG}$ | 10 |
| BAY kernel points per VG | 49 |

Ctrrot.: counter-rotating. CD: common-down





**Table 2.** Overview of the computational cases

| Case | blade pitch $\gamma$ [°] | $z_{rel,VG22}$ [-] | $z_{rel,VG16}$ [-] | $z_{rel,VG11}$ [-] | Rev. computed | $n_{\Delta t}$ per rev. [-] |
|---|---|---|---|---|---|---|
| DesNoVG | 2.53 | - | - | - | 4 | 720 |
| DesVGout | 2.53 | - | [0.15; 0.22] | [0.22; 0.3] | 2 | 720 |
| DesVGin | 2.53 | [0.077; 0.15] | [0.15; 0.22] | [0.22; 0.3] | 2 | 720 |
| oDesNoVG | -2.47 | - | - | - | 2 | 720 |
| oDesVGout | -2.47 | - | [0.15; 0.22] | [0.22; 0.3] | 2 | 720 |
| oDesVGin | -2.47 | [0.077; 0.15] | [0.15; 0.22] | [0.22; 0.3] | 2 | 720 |

$n_{\Delta t}$: number of physical time steps; VG22, VG16 and VG11 correspond to counter-rotating common-down flow VGs with $h_{VG} = 0.022\,\mathrm{m};\ 0.016\,\mathrm{m}$ and $0.011\,\mathrm{m}$ respectively

In this work, two VG setups with a different radial starting position are considered (see Table 2). For the first setup (case VGout) the VG array starts at $z_{rel} = 0.15$ and for the second setup (case VGin) the VG array starts at the blade root (i.e. $z_{rel} = 0.077$). The motivation of these two choices is explained as follows:

It is well established that rotational augmentation increases the local lift force. As already discussed in Sect. 1,

the VG vortices increase the streamwise momentum in the lower boundary layer, the flow is less affected by separation, experiences more viscous stresses and is consequently less affected by rotational augmentation (Bangga, 2018; Troldborg et al., 2016). It is not clear if the loss of rotational augmentation can be compensated or even overcompensated by the effect of the VGs. Therefore the cases with the ending "VGin", with a fully controlled blade i.e. VGs starting from the innermost part of the blade towards $z_{rel} = 0.3$ are considered. It is also possible that the coexistence of both effects (i.e. rotational augmentation

for the innermost root section and VG vortices more outwards) leads to the best outcome. Therefore, the cases with the ending "VGout", with the innermost part of the blade $z_{rel} = [0.077; 0.15]$ left uncontrolled, are regarded.

## 2.5 Simulation Cases

In the this work and as presented in Table 2, six cases are considered. They arise from three different boundary layer control states of the suction side of the blade: an uncontrolled case (NoVGs) and two cases with VG arrays (starting at a different

radial position). Each of them is analysed for two flow states: a design condition and an off-design condition. The regarded generic HAWT was designed to operate without the use of VGs. Thus, for design conditions, the separated regions in the root area have only a small extent in radial direction. In contrast to other passive flow control devices (e.g. gurney flaps), VGs do not increase the lift over the entire range of angle of attack $\alpha$ but only delay the viscous cambering occurring at high $\alpha$. This is why an off-design case is investigated by decreasing the blade pitch by $5°$ in order to increase the effective angle of attack for

each blade section and enforce larger separation. The value has been adapted in order to maintain the chordwise VG positions in fully attached flow along the entire blade. The turbine is simulated with fully-turbulent surfaces at uniform inflow velocity $U_{ref} = 9.5\,\mathrm{ms}^{-1}$ and constant rotational speed of $n = 16.1\,\mathrm{RPM}$ for all cases. For the the dual-time stepping technique, 720 physical time steps are computed per revolution with 100 inner iterations per physical time step. All cases are initialised at first

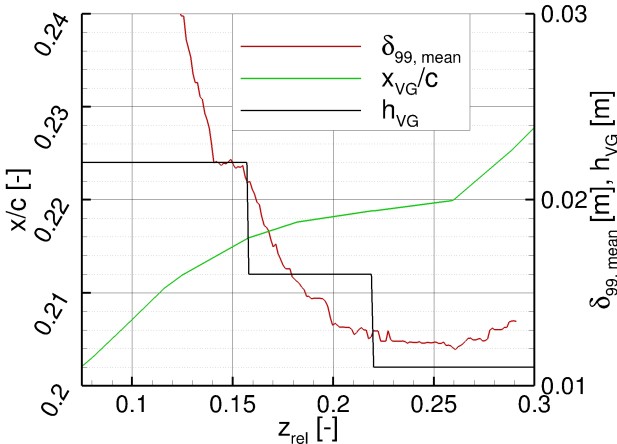

**Figure 5.** Chordwise VG positions $x_{VG}/c$, height of the VGs $h_{VG}$ and averaged boundary layer thickness $\delta_{99}$ of the uncontrolled design (DesNoVG) and off-design case (oDesNoVG) along the relative radial position $z_{rel}$.

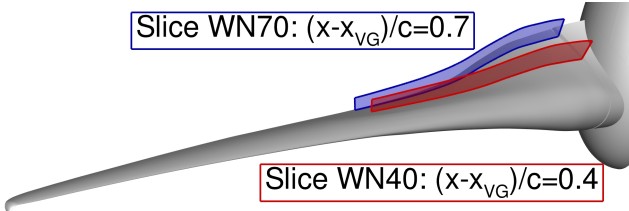

**Figure 6.** Positions of the evaluated wall-normal slices on the rotor blade. The red wall-normal slice WN40 is at 40% of the local relative chord length behind the VG array positioning line $x_{VG}/c$ and the blue wall-normal slice WN70 positioned in the same way at 70%.

with two uncontrolled revolutions. For the controlled cases two more revolutions are computed to guarantee the convergence of the BAY-model.

# 3 Results 1 - Investigation on rotational augmentation for the uncontrolled cases

As already mentioned, the intensity of the rotational augmentation is highly dependent on the regarded turbine (Herráez et al., 2014). For this reason, the turbine used in this work is at first investigated for the uncontrolled cases (DesNoVG and oDesNoVG). It was also highlighted in Sect. 1.1 that comparisons to the 2D flows can only give a rough idea of the effects due to various uncertainties. For this reason, the following results directly focus on the 3D case. Firstly, the unsteady interaction between the Himmelskamp force and the blade loads of each radial section are addressed. Secondly, an analysis of the time-averaged 3D boundary layer parameters through the shape factor $H_{12}$ is shown.

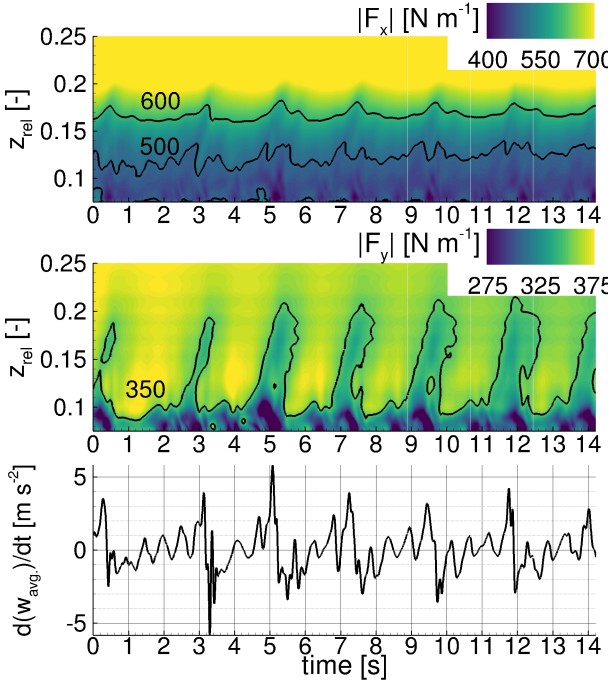

**Figure 7.** Load distribution of the uncontrolled design case (DesNoVG) along the relative radial position $z_{rel}$ over time and derivative over time of the averaged radial velocity $w_{avg}$ for the wall-normal slice WN70.

### 3.1 Blade loads and Himmelskamp force

The blade loads are extracted for each physical time step $n_{\Delta t}$ along 192 sections which are refined towards the blade root
and the blade tip. The Himmelskamp force, stated in a universal form in Eq. (1), can be greatly simplified for this work: The centrifugal force is constant for each cell in time and the Coriolis force only depends on the velocity vector in radial direction and rotation direction in the rotating frame of reference. As a consequence of the cross product in the Coriolis force ($\boldsymbol{\Omega} \times \boldsymbol{V}$), only the radial velocity $w$ has a positive effect on the state of the boundary layer by forcing a flow acceleration towards the trailing edge. Hence, in order to get an insight into the unsteadiness of the Himmelskamp force, only $w$ is regarded through the
averaged radial velocity $w_{avg}$ calculated for the wall-normal slice WN70 (blue surface in Fig. 6) extracted from the flow field at a relative chordwise position of 70% behind the VG positioning line (i.e. around 90% of the relative chord length). For each physical time step $w_{avg}$ was calculated as follows

$$w_{avg} = \frac{1}{A_{WN70}} \int w \, dA_{WN70} \tag{5}$$

where $w$ represents the velocity in radial direction for each cell over the wall-normal slice and $A_{WN70}$ represents the surface
of the wall-normal slice WN70.

In Fig. 7, the thrust force $F_x$, the driving force $F_y$ and the acceleration in radial direction (derivative in time of $w_{avg}$) as a direct indicator of the action of centrifugal force are plotted for the DesNoVG case. The variations of the forces in time for





$z_{rel} < 0.1$ are attributed to flow separation interacting in an unsteady way with the root vortices. Phases of strongly detached flow (seen by the load alleviation mostly in $F_y$ e.g. at around $5.0\,\mathrm{s}$) are followed by phases of more attached flow (load increase

e.g. $1\,\mathrm{s}$ to $1.9\,\mathrm{s}$) in a cyclic manner where the phases of highly detached flow are shorter in time than the phases of lower separation for this case.

With a frequency of around $0.5\,\mathrm{Hz}$, a similar unsteady behaviour of the flow field is visible through the load fluctuations e.g. $4.5\,\mathrm{s}$ to $5.6\,\mathrm{s}$; $6.8\,\mathrm{s}$ to $7.5\,\mathrm{s}$ and $9.0\,\mathrm{s}$ to $10.0\,\mathrm{s}$. At first, the flow separates over a large extent in radial direction which is visible through the driving force alleviation in the root section for $z_{rel} < 0.1$. In this flow state, the viscous stresses in the

boundary layer are low and thus the flow is very receptive for the action of the centrifugal force transporting the separated flow in radial direction. This is visible by the high radial acceleration $dw_{avg}/dt$ of the flow and simultaneously the extended oblique area of reduced thrust $F_x$ and driving forces $F_y$ along the radial direction $z_{rel}$. The resulting radial velocity $w$ is the main driver of centrifugal pumping and the chordwise Coriolis force ($F_{Coriolis,y} = 2\,\omega_x\,w$). The latter described effect is shown in Fig. 8 through the upper snapshot at $t = 5.4\,\mathrm{s}$ which corresponds to a cycle of increasing rotational effects (green

cycle). As soon as the separation reduces in size, shown in the lower snapshot of Fig. 8 at $t = 5.8\,\mathrm{s}$, the viscous stresses in the boundary layer increase and the impact of the centrifugal force is damped. Consequently $w$, the main driver for centrifugal pumping and the Coriolis force, reduces (i.e. deceleration). This finally helps separated flow to emerge and grow again in the root area promoting a new cycle of increasing rotational effects. The results for the DesNoVG show that large flow separations with a certain critical extend in radial direction are the starting point of a cycle increasing of rotational effect (green cycle in

Fig. 8).

Furthermore, a root vortex system composed of two counter-rotating vortices is visible on the rearmost wall-normal slice (Fig. 8 at $t = 5.8\,\mathrm{s}$). This system produces a momentum transfer towards the blade surface and is therefore responsible for the increase in driving force in Fig. 8 at $z_{rel} \approx 0.125$ which occurs after each of the three mentioned time intervals and is thus part of the cyclic behaviour of rotational augmentation for this case.

In Fig. 9 the loads for the uncontrolled off-design case oDesNoVG are shown. With respect to the transport of the detached flow outwards, the case shows similarities with the DesNoVG case considered before. For $z_{rel} > 0.12$ alternating diagonal areas of higher and lower driving force $F_y$ are visible. The phases of strong radial acceleration and deceleration of the flow occur during the periods of decreasing and increasing $F_y$ respectively. In this time interval a cycle of increasing rotational effects is followed by a cycle of decreasing rotational effects according to Fig. 8.

Regarding the inner part of the blade (around $z_{rel} = 0.1$), the driving force is at a high level over the entire time compared to the outer sections. This observation is largely different compared to the previous case (DesNoVG) and attributed to two effects: Firstly, the more efficient and more time-constant centrifugal pumping mechanism inhibits the growth of separated flow structures for $z_{rel} < 0.1$. Secondly, enabled by the strong centrifugal pumping and thus low extent of separated regions, the root vortex system (composed of two counter-rotating vortices (Bangga, 2018)) is closer to the surface compared to the

DesNoVG case and thus locally reenergizes the boundary layer resulting in locally increased blade loads ($z_{rel} \approx 0.09$).





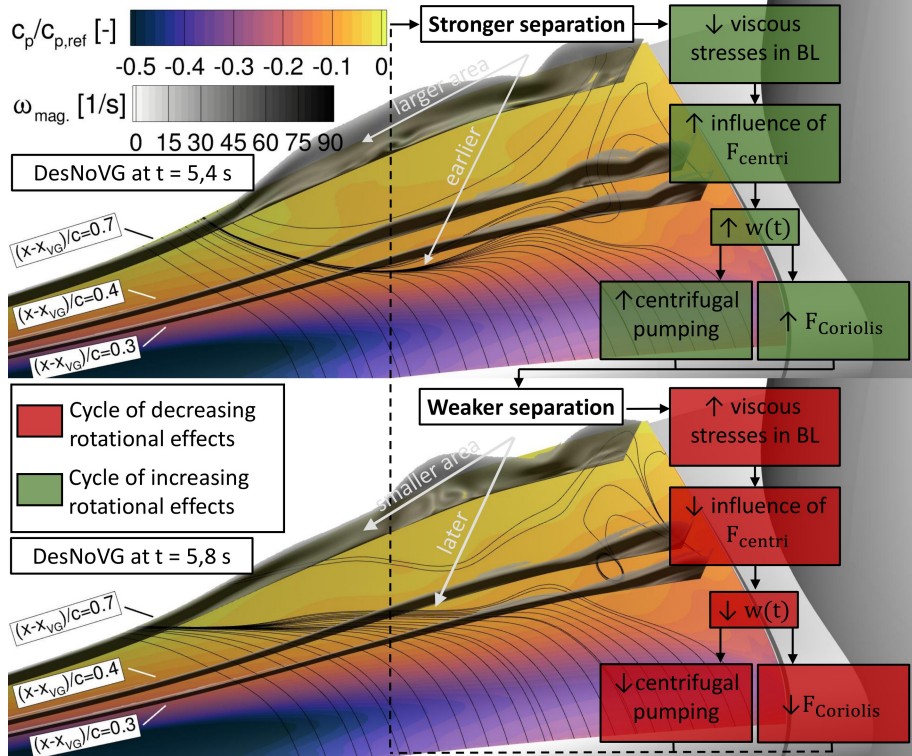

**Figure 8.** Description of the periodic behaviour of rotational augmentation in the uncontrolled cases with snapshots of the flow field at different times. BL: Boundary Layer, w(t): radial velocity.

## 3.2 Assessment of the boundary layer state

The investigation of the blade loads is not sufficient to understand the flow mechanisms and characterise the state of the boundary layer flow. In order to get a more detailed insight into the boundary layer, this section addresses the behaviour of the shape factor $H_{12}$ which is defined as

$$H_{12} = \frac{\delta_1}{\delta_2} \tag{6}$$

where $\delta_1$ is the displacement thickness and $\delta_2$ the momentum thickness of the boundary layer. Low values of $H_{12}$ indicate a fully attached boundary layer and, according to Castillo et al. (2004), values above $H_{12} \approx 2.76 \pm 0.23$ are obtained for detached boundary layers in turbulent flow. At this point it should be mentioned that both thicknesses are computed with the chordwise and spanwise velocity along wall-normal lines and consequently $H_{12}$ gives information about the state of the 3D boundary layer. This is important because the spanwise component plays a crucial role for the quantification of the influence of the rotational augmentation on the state of the boundary layer. The boundary layer thickness $\delta_{99}$, required to calculate $\delta_1$ and

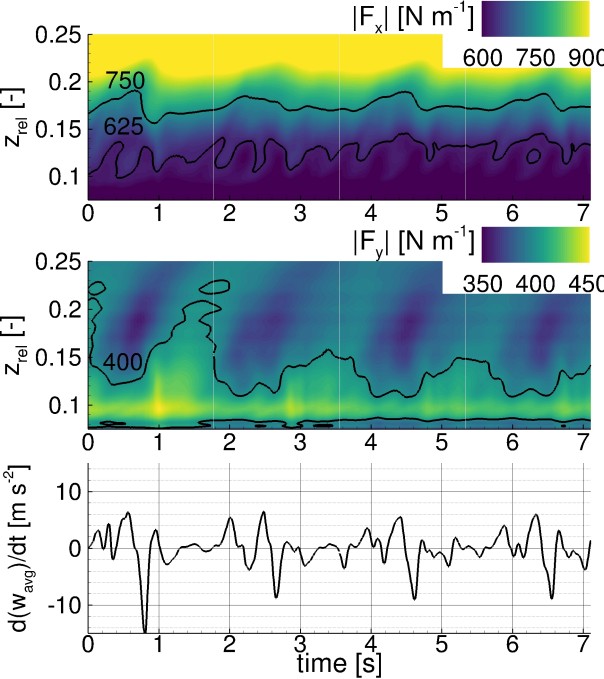

**Figure 9.** Load distribution of the off-design case (oDesNoVG) along the relative radial position $z_{rel}$ over time and derivative over time of the integrated radial velocity $w_{avg}$ for the wall-normal slice WN70.

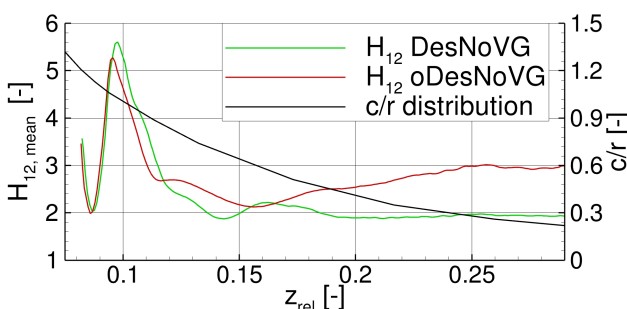

**Figure 10.** Mean distribution of shape factor $H_{12}$ for the DesNoVG and the oDesNoVG cases and chord to radius distribution ($c/r$) along the relative radial position $z_{rel}$ on the wall-normal slice WN40.

$\delta_2$, was determined through a threshold combination of the turbulence kinetic energy $k_t < 0.001\,\mathrm{m^2s^{-2}}$ and the wall distance $dw > 0.001\,\mathrm{m}$.

In Fig. 10 $H_{12}$ along the wall-normal slice WN40 (red surface in Fig. 6) and the chord length to radius distribution $c/r$ are plotted for both uncontrolled cases. In the inner blade section at $z_{rel} < 0.12$ large radial variations of $H_{12}$ are visible for both cases which is attributed to the action of the counter-rotating root vortex system. Such a system introduces high wall-normal



velocities leading to attached or detached flow depending on the direction of rotation and the radial position of the vortices and the distance between the two vortex cores.

Interestingly, the $H_{12}$ values of oDesNoVG show no offset to a higher value compared to DesNoVG even with the effective
angle of attack being larger by $5°$ (decreased pitch). Regarding the local maximum at $z_{rel} \approx 0.095$, the oDesNoVG case is even lower and of smaller radial extent. This is an indication that, as already seen in the loads, the rotational effects are larger for the oDesNoVG case. Further towards to the blade tip, the effect of rotational augmentation is alleviated due to the decreasing $c/r$ ratio and thus $H_{12}$ increases for the oDesNoVG case as one could expect for a non-rotating airfoil section.

In this section it has been shown that, in regard to the DesNoVG case, rotational augmentation on the investigated rotor blade
is not constant but occurs in cycles of increasing and decreasing rotational effects. The recurrence of the cycles is supposed to be mainly dependent on the extent of areas of separated flow in the inner blade region. For the oDesNoVG case, the innermost part of the blade encounters relatively high loads resulting from strong centrifugal pumping and from the root vortex system whereas for $z_{rel} > 0.12$ alternating phases of higher and lower loads are visible. When comparing both cases, it was confirmed through $H_{12}$ that rotational augmentation is much stronger for the oDesNoVG case than for the DesNoVG case as the states of
the respective boundary layers are on a similar level for the root section ($z_{rel} < 0.16$).

## 4    Results 2 - Investigations of the blade root equipped partially with VGs

In the first result chapter it was shown that the inner part of the blade is affected by rotational augmentation effects of variable extent. In more detail, the results of the oDesNoVG case reveal high driving forces for the inner blade root section through the positive effect of strong centrifugal pumping. With the aim of using this beneficial effect on the blade root, in the following
section, the blade is only partially equipped with VGs for $z_{rel} = [0.15; 0.3]$. The radial starting location of the controlled section was determined based on the observation in Sect. 3.2 where only low rotational effects above $z_{rel} \approx 0.15$ were observed.

### 4.1    Blade loads and Himmelskamp force

In Fig. 11 the VG impact on the time-averaged loads is plotted for all six cases considered in this work. For the DesVGout case,
the loads are equal or slightly lower than for the uncontrolled case depending on the radial position. Globally, the differences

**Table 3.** Thrust force $F_x$ and torque $M_y$ comparison for $z_{rel} < 0.3$ of the different VG cases to the uncontrolled case.

| Case | $\Delta F_{x,Case-NoVG}$ [%] | $\Delta M_{y,Case-NoVG}$ [%] |
|------|------|------|
| DesVGout | -1.18 | -2.05 |
| oDesVGout | 1.77 | 2.83 |
| DesVGin | 0.75 | -0.20 |
| oDesVGin | 3.58 | 3.28 |



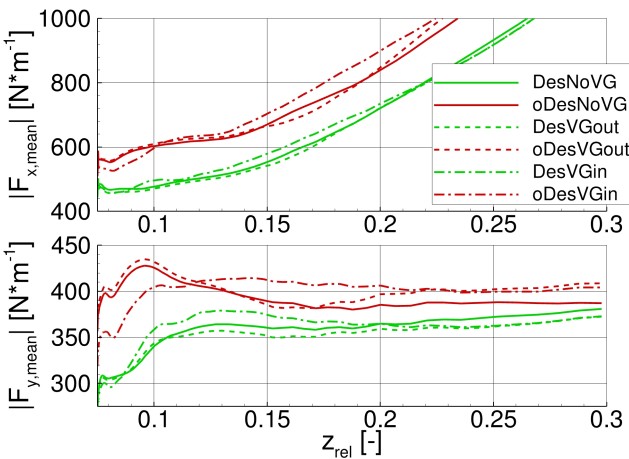

**Figure 11.** time-averaged load distribution of thrust force $F_x$ and driving force $F_y$ for all regarded cases along the relative radial position $z_{rel}$.

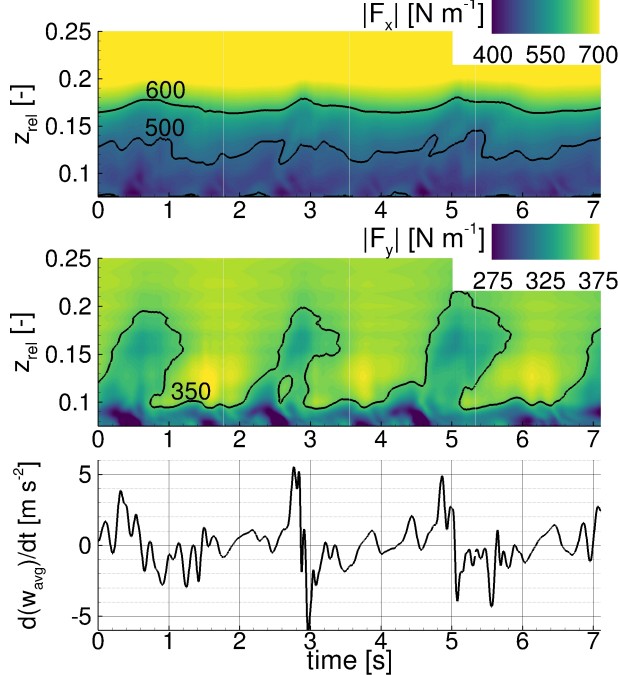

**Figure 12.** Load distribution of the design case partially equipped with VGs for $z_{rel} = [0.15; 0.3]$ (DesVGout) along the relative radial position $z_{rel}$ over time and derivative over time of the integrated radial velocity $w_{avg}$ for the wall-normal slice WN70.

in torque $\Delta M_{y,DesVGout-NoVG}$ for the inner section ($z_{rel} < 0.3$) is lower by 2.05% (Table 3) compared to the case without VGs. The reduction is due to the lower driving force $F_{y,mean}$ for $z_{rel} > 0.11$ (Fig. 11). In this region and for this case, the





flow is fully attached and consequently the VGs do not have any benefit, even worse, they inhibit any lift enhancement through radial flow which is presumed to be the reason for the lower driving force. In contrast to the VG mechanism, the radial flow

resulting from rotational augmentation also increases the lift at moderate angles of attack (Herraéz et al., 2016) and does not only delay flow separation for higher angles of attack.

Regarding the unsteady loads in Fig. 12, similar phases of increasing and decreasing rotational effects as for the DesNoVG case are visible by the areas of lower driving force traveling outwards in time. The phases of radial acceleration instantaneously follow the occurrence of large detached areas in the root area ($z_{rel} < 0.1$). Thus, the VGs placed partially on the blade (at

$z_{rel} > 0.15$) do not inhibit the occurrence of cyclic rotational augmentation effects in the root area.

For the oDesVGout case, the time-averaged driving force $F_y$ is slightly higher in the inner part of the blade than for the uncontrolled case. This again shows that the rotational effects in this case are not affected by the VGs. In contrast to the DesVGout case, the benefit of the VGs is well visible for $z_{rel} > 0.2$ through higher time-averaged thrust and driving force compared to the uncontrolled case (11). At those sections, the impact on torque is high, leading to a 2.83% increase compared to

the uncontrolled case (Table 3). This behaviour was expected because the considered turbine was designed to be used without VGs and for lower effective angles of attack. Remarkably, the VGs are not efficient (i.e. increasing the loads) from their radial start position at $z_{rel} = 0.15$ but only for $z_{rel} > 0.18$. This is attributed to the interaction of the centrifugal pumping mechanism in radial direction and the VG vortices convecting in chordwise direction. Explained in more detail, the centrifugal pumping behaves in an undisturbed manner for $z_{rel} < 0.15$ (separated flow is accelerated towards the blade tip) and impacts

perpendicularly the chordwise propagating VG vortices only at $z_{rel} = 0.15$. Therefore, the collected radial momentum of the separated flow during the acceleration is progressively reduced but still strong enough to reduce the efficiency of the first VG vortices.

In Fig. 13, the loads of the oDesVGout case are plotted over time in the same manner as in the previous section. The unsteady behaviour is very similar to the oDesNoVG case and thus the rotational augmentation effect is not diminished by the VGs. One

difference due to the VGs is the occurrence of high frequency fluctuations in the radial acceleration $dw_{avg}/dt$ resulting from interaction of spanwise flow and chordwise VG vortices.

## 4.2 Assessment of the boundary layer state

In Fig. 14 the shape factor $H_{12}$ of the 3D boundary layer is plotted. The root section displays results similar to those of the uncontrolled case already described in Sect. 3.2, i.e. rotational augmentation is the dominant flow mechanism. The VGs are

clearly visible through the zigzag pattern of $H_{12}$: counter-rotating common-down flow VG pairs create downwash areas between each other and upwash areas between them and their respective neighbour vortices from the other pair. In the downwash areas, the velocity vectors are partially deflected towards the surface and therefore reduce $H_{12}$. In contrast, the upwash areas deflect the flow away from the wall and yield a higher $H_{12}$. Regarding the second VG array VG11 ($z_{rel} > 0.22$ in Fig. 14) , an offset between the two cases is visible due to the higher effective angles of attack. This offset is not visible for the first VG

array VG16 at $0.2 < z_{rel} < 0.22$ and underlines the importance of the correct adaptation of the VG height to the local state



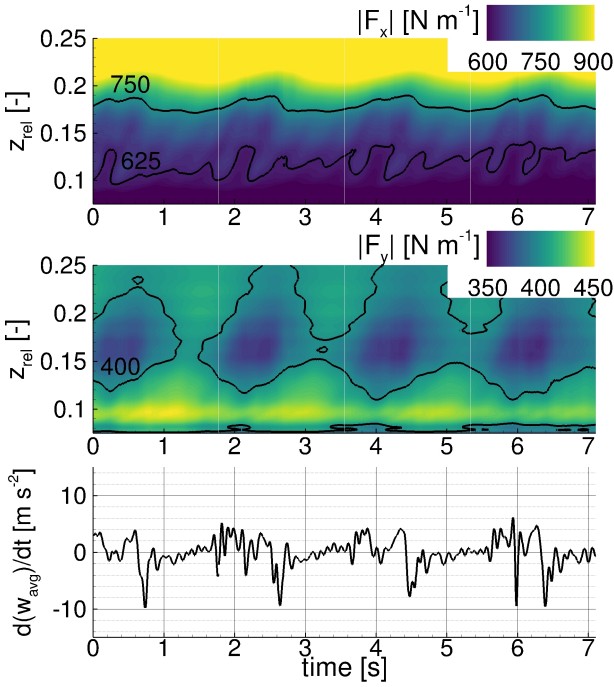

**Figure 13.** Load distribution of the off-design case partially equipped with VGs for $z_{rel} = [0.15; 0.3]$ (oDesVGout) along the relative radial position $z_{rel}$ over time and derivative over time of the integrated radial velocity $w_{avg}$ for the wall-normal slice WN70.

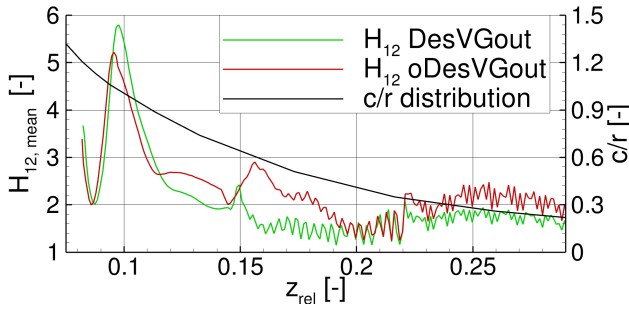

**Figure 14.** Mean distribution of shape factor $H_{12}$ on the wall-normal slice WN40 for the partially controlled cases with VGs for $z_{rel} = [0.15; 0.3]$ (DesVGout and oDesVGout) and chord to radius distribution ($c/r$) along the relative radial position $z_{rel}$.

of the boundary layer. A low height can lead to an early separation whereas an oversized height creates additional drag without further benefit on the lift.

As already seen for the time-averaged blade loads, the impact of the VGs is not visible in $H_{12}$ from the first VG of the array, which starts at $z_{rel} > 0.15$, but only from $z_{rel} > 0.16$ for DesVGout and from $z_{rel} > 0.18$ for oDesVGout. Furthermore, Fig. 14 reveals a local peak in $H_{12}$ at $z_{rel} \approx 0.16$ for oDesVGout and a much smaller one in amplitude and spanwise extent




WIND
ENERGY
SCIENCE
DISCUSSIONS
european academy of wind energy
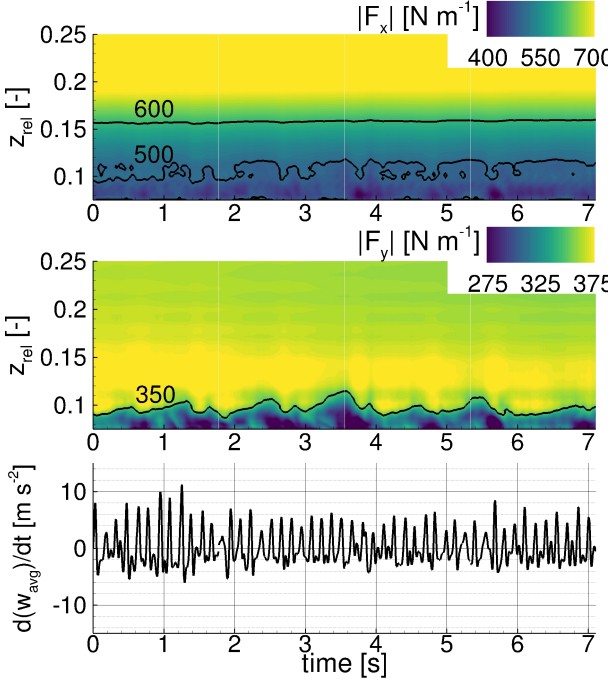

**Figure 15.** Load distribution of the design case entirely equipped with VGs for $z_{rel} = [0.077; 0.3]$ (DesVGin) along the relative radial position $z_{rel}$ over time and derivative over time of the integrated radial velocity $w_{avg}$ for the wall-normal slice WN70.

at $z_{rel} \approx 0.15$ for DesVGout. In both cases, the peak results from the same mechanism, i.e. the negative interaction of centrifugal pumping (radial flow) and the VG vortices (chordwise flow). The extent of this interaction depends on the intensity of centrifugal pumping, which is much stronger for the oDesVGout case and the VG vortex strength which is similar for both cases (because it mainly depends on the VG size). Hence, for the oDesVGout case a longer radial length is required to reduce

the higher spanwise momentum. It is supposed that a larger VG height for this case could further increase its performance by stopping abruptly the spanwise flow and enabling fast takeover of 2D VG controlled flow. Nevertheless, the consequences of stronger VG vortices on the separation of the boundary layer should be evaluated in more in detail.

## 5 Results 3 - Investigations of the blade root equipped entirely with VGs

In this chapter, the blade root is fully equipped with VGs along $z_{rel} = [0.077; 0.3]$ starting from the innermost relative radial

position of the blade mesh. At this point, it should be mentioned that the connector (light blue in Fig. 3) is left uncontrolled. This VG setup aims to give a quantitative comparison between the exclusive effect of VGs and the exclusive effect of rotational augmentation (i.e. the uncontrolled cases).



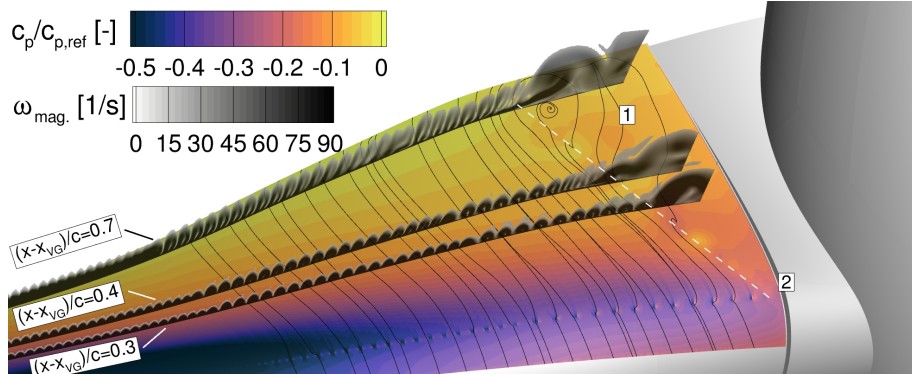

**Figure 16.** Pressure distribution on the suction side and the vorticity magnitude $\omega_{mag}$ for the DesVGin case at $t = 3.6\,\mathrm{s}$. The slices are placed in wall-normal direction and at different constant relative chordwise positions behind the VGs $(x - x_{VG})/c$.

## 5.1 Blade loads and Himmelskamp force

In Fig. 11 the time-averaged loads are plotted for the DesVGin case. In the inner part of the blade ($z_{rel} < 0.2$) the loads are
globally increased compared to the two other identically pitched cases (DesNoVG and DesVGout). It has already been shown for those cases that the inner part of the blade encounters flow separation but the centrifugal pumping is not very efficient. Thus, the benefit of VGs in the inner blade area is higher than the one resulting from rotational augmentation. Despite the favourable effect for the inner section, the torque $M_y$ of the entire section ($z_{rel} < 0.3$) is almost equal to the uncontrolled case (Table 3). This is due to the slightly decreased driving force for $z_{rel} > 0.21$. The reasons for this are the same as those already
discussed in Sect. 4.1 for DesVGout case and are mainly related to the inhibition of spanwise flow through the VG vortices and meanwhile no positive effect of the VGs.

For the load fluctuations shown in Fig. 15 two different aspects are discussed: Firstly, the separated flow visible for $z_{rel} < 0.1$ and secondly, the inhibition of the spanwise transport of separated structures.

Regarding the first aspect, separated flow for $z_{rel} < 0.1$ develops even if VGs are placed on the entire area. This is due to the
highly outward directed (i.e. skewed) flow combined with the blocking effect of spanwise flow through the VG vortices which enables the formation of a pocket of separated flow. In order to get qualitative insight, a snapshot of the flow field at $t = 3.6\,\mathrm{s}$ is displayed in Fig. 16. The interaction of the separated flow and the first VG vortices creates a shear layer highlighted by the white dashed line. The dynamic of it is visible through the high frequency oscillations of the acceleration of $w_{avg}$. Furthermore, it is clearly visible through the streamlines upstream of the VG arrays that the VGs encounter highly skewed inflow resulting in
high inclination angles for each the VG of a pair located closer to the hub and at the same time lower or no inclination angle of the other VG of the pair (located closer to the blade tip). Those inflow conditions are far from ideal because the quality of the shed vortex system is degraded (Pearcy, 1961). Thus, a rotation of the innermost VG pairs around their symmetry center may help to avoid the misalignment. Finally, the use of wing-type vortex generators for these sections may be beneficial as they are mainly insensitive to skewed inflow conditions (Pearcy, 1961).



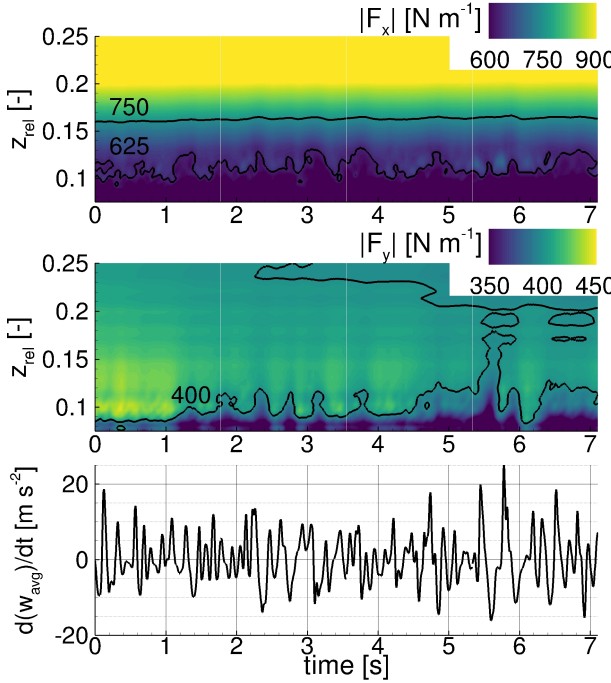

**Figure 17.** Load distribution of the off-design case entirely equipped with VGs for $z_{rel} = [0.077; 0.3]$ (oDesVGin) along the relative radial position $z_{rel}$ over time and derivative over time of the integrated radial velocity $w_{avg}$ for the wall-normal slice WN70.

Regarding the second aspect, in contrast to the already considered identically pitched cases, the transport of separated flow in spanwise direction is inhibited and the loads are almost constant over time for $z_{rel} > 0.12$. Even for large separated flow structures (e.g. $t = 3.6$ s in Fig. 15), an oblique area of outward transport of the separated flow is visible. Thus, the centrifugal force is not able to accelerate the flow sufficiently in spanwise direction to lift-off a large number of VG vortices. Nevertheless, the spanwise acceleration $w_{avg}$ over the regarded slice is oscillating in a higher frequency than for the other considered cases

which is related to the cyclic interaction of the inner VG vortices and a small centrifugal pumping effect of the detached flow.

For the oDesVGin case, the time-averaged loads are plotted in Fig. 11. The centrifugal pumping mechanism is largely reduced and consequently the local maximum in $F_y$ at $z_{rel} \approx 0.09$ is not visible for this case. In return, for $z_{rel} > 0.115$ the loads are higher than for the other two cases with the same pitch (oDesNoVG and oDesVGout) which is due to the reduced negative interaction of centrifugal pumping with the VG vortices increasing greatly their efficiency. The inner parts are much

less relevant than the outer ones for the torque $M_y$ and therefore this case is reaching a 3.28% higher value than for the NoVG case (Table 3).

The loads over time for the oDesVGin case are plotted in Fig. 17. At the beginning of the time series until 1 s, high values of driving force are visible around $z_{rel} = 0.1$ and the separated areas are weak. After this time interval, larger and stronger separated areas appear. At this point it should be mentioned that this is probably not a result of insufficient numerical con-

vergence because all evaluated time intervals in this work are based on a minimum of two revolutions with the uncontrolled



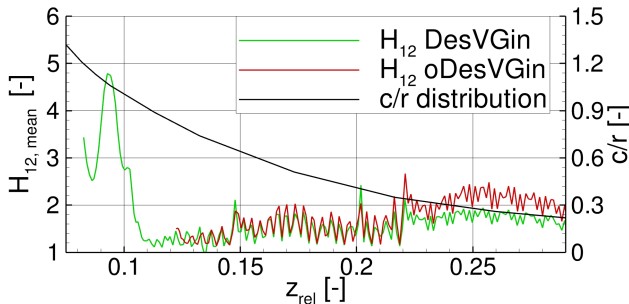

**Figure 18.** Mean distribution of shape factor $H_{12}$ of the wall-normal slice WN40 for the fully controlled cases with VGs (DesVGin and oDesVGin) and chord to radius distribution $(c/r)$ along the relative radial position $z_{rel}$.

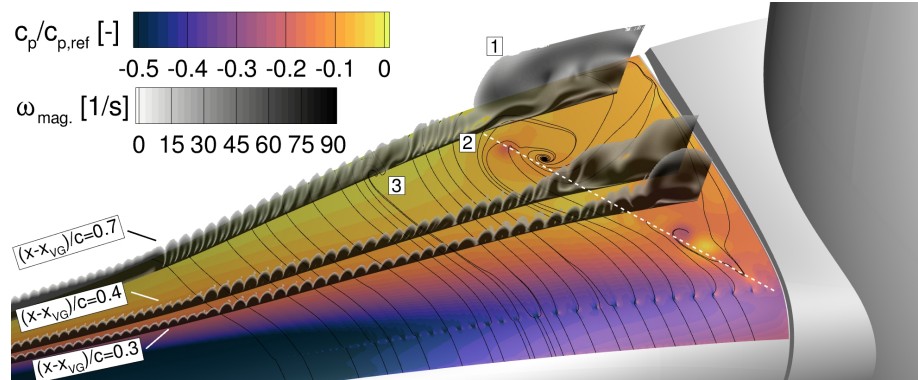

**Figure 19.** Pressure distribution on the suction side and the vorticity magnitude $\omega_{mag}$ for the oDesVGin case at $t = 3.6\,\mathrm{s}$. The slices are placed in wall-normal direction and at different constant relative chordwise positions behind the VGs $(x - x_{VG})/c$.

settings and two revolutions with controlled settings. The results are rather supposed to be related to the bimodal behaviour of the lift signatures of airfoils equipped with VGs just after $c_{l,max}$ shown experimentally by Baldacchino et al. (2018) for a DU97-W-300 airfoil section in a wind tunnel. They observed this effect, which is resulting from a highly dynamic process of separation and reattachment only when equipped with VGs. On a rotor blade, the rotational augmentation also takes part in

the interaction as soon as separation occurs. Thus, the already explained cycles of increasing and decreasing rotational effects (Fig. 8) gets even more complex potentially affecting the fatigues loads of the blade in a new manner and should therefore be the object of further research.

## 5.2    Assessment of the boundary layer state

In Fig. 18, the shape factor $H_{12}$ is shown for the two cases DesVGin and oDesVGin. For $0.12 < z_{rel} < 0.22$ both cases are

very similar because the VGs of the VG16 array are able to provide a fully attached boundary layer for both cases (DesVGin and DesVGout). In contrast, the smaller VG11 array ($z_{rel} > 0.22$) is not able to reach the same $H_{12}$ level for both cases.



The higher pitch of the oDesVGin case leads to a still attached but deteriorated state of the boundary layer visible through the offset of $H_{12}$ to higher values. For $z_{rel} < 0.12$ the oDesVGin case shows largely separated flow, as visible in Fig. 19, and consequently it was not possible to determine $H_{12}$ because the edge of the boundary layer $\delta_{99}$ was not detectable for the extracted slice.

By having a closer look at the first innermost VG vortex pairs detected with $H_{12}$ for oDesVGin (Fig. 18) at $z_{rel} \approx 0.125$ a spanwise shift of the peaks is visible between the two cases. According to those peak positions the vortex pairs of the oDesVGin case shown in Fig. 19 by the framed "2" are pushed outwards and also stretched in wall-normal direction by the strongly separated flow (Fig. 19 framed "1") for $z_{rel} < 0.12$. Furthermore, a shear layer (dashed white line) similar to the one observed for the DesVGin case is visible but the inclination of the line is higher. This is related to the fact that for the higher pitch angle the local adverse pressure gradient is stronger and relocates the loss of efficiency of the VG vortices to a more upstream chordwise position. Due to this same effect, a small recirculation area is visible at the spanwise transition to a smaller VG height (framed "3"). At this point it is mentioned that the spacing at these transitions was set to the interspacing of the larger VGs. Additional studies on an extruded airfoil have shown that this choice of spacing is less prone to separation because the stronger vortex obtains more space to evolve in spanwise direction and the lift-off from the surface (i.e. loss of efficiency) through the induction of the VG vortex of the smaller array is delayed.

## 6  Conclusions

In this work, a rotor blade of a generic $2\,\mathrm{MW}$ wind turbine was investigated by means of unsteady RANS simulations with the CFD solver FLOWer regarding rotational augmentation and its interaction with vortex generators. In order to reduce the required computational resources, a $120°$-model was used. The suction side of the blade has a very high mesh resolution (83.9 million cells) to resolve the small VG vortices. To further reduce the computational effort, a BAY-type model was used to model the parabolically-shaped VGs considered in this study. The placement of the VGs and their size was adapted according to current best-practices. Regarding the flow conditions of the blade, two operational conditions were studied. The first one is a given design case and the second one is an off-design case where the blade pitch angle was adapted to generate flow separation within a certain distance behind the chordwise VG position. These two operating conditions were investigated at first for the uncontrolled case and then with two different VG setups. For the first one, the inner part of the blade is left uncontrolled and the VGs are placed only further outwards ($0.15 < z_{rel} < 0.3$) to investigate the interaction of the rotational augmentation and the VG vortices. For the second one, the VGs are placed all over the root region of the blade ($0.077 < z_{rel} < 0.3$) to minimize any kind of separated flow.

The cases were investigated regarding the torque and the thrust force over the inner blade part as well as the time-averaged loads. In addition, the time-resolved loads were examined and compared to the time-resolved radial flow acceleration which is an indicator of the action of the centrifugal force being a main driver of rotational augmentation. Finally, in order to get a deeper insight into the state of the boundary layer, the shape factor $H_{12}$ was studied in the wake of the VG arrays.

Regarding the stated scientific questions in Sect. 1.3, the findings in this work allow the following answers:

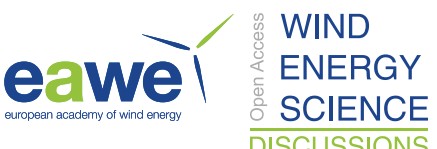

Q1: For the uncontrolled cases the rotational augmentation for the off-design case is far more efficient than for the design case (with lower effective angles of attack). The constant centrifugal pumping over time leads to a load peak in the innermost root region of the blade. For the design case the centrifugal pumping appears to be a cyclic phenomenon with increasing and decreasing rotational effects recurrent in time.

Q2: The combination of rotational augmentation in the inner blade section ($z_{rel} < 0.15$) and the effect of VGs further outwards ($0.15 < z_{rel} < 0.3$) is rather destructive for the regarded cases and the considered turbine. Presumably, this is also true in general because the driver of the centrifugal pumping is the radial flow which is undermined by the VG vortices convecting in chordwise direction. Hence, VG vortices alleviate the rotational augmentation effect and vice versa, the rotational augmentation locally reduces the positive VG effect by impacting the first VG vortices in radial direction, thus reducing their efficiency. Nevertheless, VGs are more efficient than rotational augmentation effects for largely separated flow (off-design case) particularly if the separation has a large radial extent where rotational augmentation effects vanish progressively. On the other hand, if the flow is mainly attached (design case), VGs are useless and even have a slight negative effect as they inhibit radial flow.

Q3: Rotational augmentation is a cyclic phenomenon for the design case and the regarded turbine. The interaction of the first VG vortices in the root region creates a shear layer which produces high frequency fluctuations in the spanwise flow.

In summary, this study highlighted the importance of not only optimizing the VGs for the relevant airfoil at the regarded blade section and inflow conditions but shows that the 3D effects occurring on a rotating blade interact with the VGs in a complex manner. To incorporate the 3D effects, the use of CFD simulations seems to be a very helpful and relatively inexpensive tool compared to experimental setups including rotating blades and VGs.

*Code availability.* FLOWer is proprietary software of DLR and has been expanded at IAG. Information can be obtained from the corresponding author.

*Author contributions.* FS implemented the BAY model, created the CFD setup, performed the computations, did the post-processing, evaluated the results and wrote the paper. TL and EK initiated the research, supervised the work and revised the manuscript.

*Competing interests.* The authors declare that they have no conflict of interest.

*Acknowledgements.* The authors are very grateful to the German Federal Ministry for Economic Affairs and Energy (BMWi) for funding the research within the framework of the joint research projects Schall_KoGe (FKZ 0324337C) and IndiAnaWind (FKZ 0325719F). The authors



gratefully acknowledge the High-Performance Computing Center Stuttgart (HLRS) for providing computational resources within the project WEALoads. Further acknowledged are Florian Wenz for providing the blade mesh and Lando Blazejewski for writing a routine to define the wall-normal field slices.





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
