# Peer review of "Numerical Study of the Unsteady Blade Root Aerodynamics of a 2MW Wind Turbine Equipped With Vortex Generators"

_Wind Energy Science, 2023_

## Referee Comment (RC1)

**General comments:**
This manuscript is a significant contribution to the understanding of the flow mechanisms governing the effectiveness of VGs on a wind turbine rotor. Generally, the manuscript is clearly written and makes proper reference to related works. The authors use high-fidelity CFD for the investigations, which is a suitable choice because it allows a detailed one-to-one analysis of the impact of different VG configurations on the inboard blade aerodynamics of a wind turbine in comparison to the case where the turbine operates without VG's. The numerical setup is well described.
However, I think the structure of the results section could be improved. Currently, this section initially describes the results for the uncontrolled case, then the case with blades partially equipped with VG's and finally the fully equipped blades. I believe that the structure would be improved by making one "Blade loads and Himmelskamp force" section, in which these are described and compared for all three cases are in a more direct manner. The same thing goes for the "Assessment of boundary layer state" section. With this restructuring, the reader would not have to go back and forth in the manuscript to compare results/figures. For example, figures 10, 14 and 18 could be gathered into one figure with three subplots and be shown as part of the boundary layer state section. At the same time, I think that you could avoid some of the repetition, which is currently in the manuscript and thereby make it shorter. Based on this, I recommend the manuscript for publications provided it be subject to minor revisions.

**Specific comments:**
L100: On the other hand, the VG should not be too low (so that they essentially are just roughness elements).
L140-146: I suggest rewording research question Q1 so that it is more general than just aimed at the considered turbine. For example to something like this: "How are the loads and the state of the boundary layer of a turbine without VGs affected by rotational augmentation for different pitch settings?" It is true that you only consider one turbine and as such cannot answer it in general but on the other hand, your analysis of the loads and boundary layer state is quite general. Although the results will not be identical for another rotor, we must expect that many of the mechanisms you show are general for all rotors.
L157-158: I think it is stretching it too much to consider these effects negligible. They could very well have a significant impact. However, from a research point of view, it is perfectly valid to neglect these effects and only focus on the rotational effects.
L188: It is not entirely clear what the size of epsilon is. You refer to Seel et al. but it would be good to state e.g. how epsilon relates to e.g. the grid spacing. It seems that epsilon is governed by the distance between the kernel points used to define the VG shape but I guess it must also depend on the spacing in the CFD grid. Could you please clarify?
L195: Please state the used values for the radial, upstream and downstream distances, respectively and not only refer to Sayed et al.
L228-230: The variation of hVG is stated to be a best fit to recommendations from previous research works, i.e. hVG<0.5delta99 and hVG<0.1c. However, Fig. 5 (and Table 1) shows that hVG ~ delta99 and from Fig. 5 it is not possible to see how hVG relates to the chord. So please clarify in the text or update Fig. 5.
Fig. 5: The caption text is a bit confusing. When I first read it, I expected two curves for the boundary layer height: one for the design case and one for the off-design case. I do understand what you mean but I still suggest rewording.
Fig. 12, 13 and 17: Why are you showing the radial flow acceleration and not just the velocity? As you write in L267-268, it is the radial velocity, which through the Coriolis force causes an acceleration towards the trailing edge. Therefore, I think it would make more sense to show the radial velocity.

L321-323: Please clarify the threshold used to define the boundary layer thickness. I don't understand the latter criterion of dw>0.0001m.

Results: I think your manuscript would benefit from a restructuring as described in the "general comments". This would more naturally facilitate a comparison of the flow state and loads with and without VGs. I understand that this may be a bit cumbersome but I believe that it would strengthen the manuscript so at least you should consider it.

**Technical corrections:**

The authors often use interposed and/or parenthetical sentences, which are not always helpful for the overall understanding. I suggest that you revisit the manuscript and decide whether these sentences are always necessary or whether they could be reworded. Below, I have made suggestions to rewording a few of them in a way that I find more readable. There are more of these sentences, which I think could benefit from a rewording but admittedly I'm not an English expert so maybe it is just me.

L243: I suggest rewording to: "This work considers six cases as presented in Table 2."

L331: Maybe rephrasing to: "This is an indication that the rotational effects are larger for the oDESNoVG case as was also seen in the blade loads."

L399: Change "chapter" to "section"

L420: The sentence "…for each the VG of a pair …. at the same time lower or no inclination angle …." is confusing. Please rephrase.

L447: Change "object" to "subject"

---

## Referee Comment (RC3)

[referee-annotated manuscript omitted]

---

## Author Comment (AC1)

**Reply to comments by Reviewer 1**

Ferdinand Seel on behalf of the authors
IAG, University of Stuttgart

June 23, 2023

The authors would like to thank the reviewer for his/her efforts and valuable comments. They are very much appreciated and incorporated into the revised paper.

The structure of the result part of the paper was revised according to the general comments of the reviewer: The results are now divided of in two parts "Blade loads and Himmelskamp force" and "Assessment of boundary layer state" and the figures will be brought closer together in the final version of the paper (double column format). All comments have been addressed and are included (if required) in the marked manuscript below.

In the present document, the specific comments given by the first reviewer are addressed consecutively. The following formatting is chosen:

- The reviewer comments are marked in blue and italic.

- The reply by the authors is in black color

- A marked-up manuscript is added. Changed sections with regard to the comments by reviewer 1 are marked in yellow. General changes made by the authors are marked in green.

**Specific comments "S"**

1. "*L100: On the other hand, the VG should not be too low (so that they essentially are just roughness elements).*"

The authors rephrased the sentence and included the proposition (see $\boxed{\text{R1:S1}}$ (page 4, line 102)).

2. "*L140-146: I suggest rewording research question Q1 so that it is more general than just aimed at the considered turbine. For example to something like this: "How are the loads and the state of the boundary layer of a turbine without VGs affected by rotational augmentation for different pitch settings?" It is true that you only consider one turbine and as such cannot answer it in general but on the other hand, your analysis of the loads and boundary layer state is quite general. Although the results will not be identical for another rotor, we must expect that many of the mechanisms you show are general for all rotors.*"

The authors did not investigate other turbines regarding rotational augmentation for now and as Herráez et al., 2014 mentioned, the effects do heavily depend on the turbine. For this reason, the authors would prefer to maintain the present wording of question Q1.

3. "*L157-158: I think it is stretching it too much to consider these effects negligible. They could very well have a significant impact. However, from a research point of view, it is perfectly valid to neglect these effects and only focus on the rotational effects.*"

The authors rephrased the sentence and included the proposition (see $\boxed{\textbf{R1:S3}}$ (page 6, line 169)).

4. "*L188: It is not entirely clear what the size of epsilon is. You refer to Seel et al. but it would be good to state e.g. how epsilon relates to e.g. the grid spacing. It seems that epsilon is governed by the distance between the kernel points used to define the VG shape but I guess it must also depend on the spacing in the CFD grid. Could you please clarify?*"

The authors stopped voluntarily at the detailed description of $\varepsilon$ as it is quite long and would require at least two other equations and their respective description.

As clarification: $\varepsilon_{i,j}$ is defined for reach kernel point in order to get an isotropic distribution of the forces over the whole virtual VG surface. It depends on the mean distance to the neighbouring kernel points (computed automatically for each kernel point) and on a single user input, which defines the isotropic smearing width of each kernel in a dimensionless way depending on the given point cloud input. The overall force will be the same for all cases, but with a high mesh resolution and a low smearing constant, the model is able to represent complex 2D VG shapes in a precise manner.

In the aim of shortening the paper, the authors would like to avoid additional descriptions of the BAY model and instead use the reference (Seel et al. (2021)) to the very detailed paper which is openly accessible for any reader.

5. "*L195: Please state the used values for the radial, upstream and downstream distances, respectively and not only refer to Sayed et al.*"

The authors added the information (see $\boxed{\textbf{R1:S5-a}}$ (page 8, line 208) and $\boxed{\textbf{R1:S5-b}}$ (page 8, line 209)).

6. "*L228-230: The variation of hVG is stated to be a best fit to recommendations from previous research works, i.e. hVG<0.5delta99 and hVG<0.1c. However, Fig. 5 (and Table 1) shows that hVG delta99 and from Fig. 5 it is not possible to see how hVG relates to the chord. So please clarify in the text or update Fig. 5.*"

The authors updated Fig. 5 by adding $h_{VG}/c$ over $z_{rel}$ and included it in the text (see $\boxed{\textbf{R1:S6-b}}$ (page 10, line 246)) Furthermore, we found a typo ($h_V G < 0.01$ not $h_V G < 0.1$ ) and corrected it (see $\boxed{\textbf{R1:S6-a}}$ (page 10, line 246)).

7. "*Fig. 5: The caption text is a bit confusing. When I first read it, I expected two curves for the boundary layer height: one for the design case and one for the off-design case. I do understand what you mean but I still suggest rewording.*"

The authors rephrased it in the text (see $\boxed{\textbf{R1:S7}}$ (page 10, line 242)) and in the description of figure 5.

8. "*Fig. 12, 13 and 17: Why are you showing the radial flow acceleration and not just the velocity? As you write in L267-268, it is the radial velocity, which through the Coriolis force causes an acceleration towards the trailing edge. Therefore, I think it would make more sense to show the radial velocity.*"

The velocity would be an absolute integral value over the entire slice WN70, and consequently it includes a part of the flow outside the boundary layer. Thus, a physical interpretation of this absolute value is difficult and may even be misleading. Furthermore, as the paper has a strong focus on the unsteadiness of the rotational augmentation, the acceleration and deceleration

phases of the flow in spanwise direction are a good indicator for start and end points of the different cycles.

For these reasons, the authors would prefer to maintain the representation of the acceleration.

9. "*L321-323: Please clarify the threshold used to define the boundary layer thickness. I don't understand the latter criterion of dw>0.0001m.*"

The slope of $k_t$ along the wall distance $dw$ starts at $k_t(dw = 0) = 0$, grows toward a maximum and reduces again to 0 for $k_t(dw = \delta_{99})$. The mentioned criterion avoids to erroneously select the very low $k_t = 0.001 m^2 s^{-2}$ position just above to wall.

10. "*Results: I think your manuscript would benefit from a restructuring as described in the "general comments". This would more naturally facilitate a comparison of the flow state and loads with and without VGs. I understand that this may be a bit cumbersome but I believe that it would strengthen the manuscript so at least you should consider it.*"

The authors modified the structure of the paper as proposed by the reviewer. The figures for the comparison of the different cases are now much closer together, or even on one page for the final format of the paper (double column).

**Technical corrections "T"**

1. "*The authors often use interposed and/or parenthetical sentences, which are not always helpful for the overall understanding. I suggest that you revisit the manuscript and decide whether these sentences are always necessary or whether they could be reworded.* "

The authors removed parenthetical sentences (see $\boxed{\textbf{R1:T1-a}}$ (page 1, line 11), $\boxed{\textbf{R1:T1-b}}$ (page 8, line 220), $\boxed{\textbf{R1:T1-c}}$ (page 22, line 440), $\boxed{\textbf{R1:T1-d}}$ (page 23, line 469))

2. "*L243: I suggest rewording to: "This work considers six cases as presented in Table 2." *"

The sentence was adapted (see $\boxed{\textbf{R1:T2}}$ (page 11, line 262)).

3. "*L331: Maybe rephrasing to: "This is an indication that the rotational effects are larger for the oDESNoVG case as was also seen in the blade loads." *"

The sentence was adapted (see $\boxed{\textbf{R1:T3}}$ (page 22, line 441)).

4. "*L399: Change "chapter" to "section"*"

The sentence was removed due to the restructuring of the result section.

5. "*L420: The sentence "...for each the VG of a pair .... at the same time lower or no inclination angle ...." is confusing. Please rephrase.*"

The sentence was adapted (see $\boxed{\textbf{R1:T5}}$ (page 19, line 386)).

6. "*L447: Change "object" to "subject"*"

The sentence was adapted (see $\boxed{\textbf{R1:T6}}$ (page 21, line 418)).

[revised manuscript text omitted]

---

## Author Comment (AC2)

**Reply to comments by Reviewer 2**

Ferdinand Seel on behalf of the authors
IAG, University of Stuttgart

June 23, 2023

The authors would like to thank the reviewer for his/her efforts and valuable comments. They are very much appreciated and incorporated into the revised paper.

In the present document, the comments given by the second reviewer are addressed consecutively. The following formatting is chosen:

- The reviewer comments are marked in blue and italic.

- The reply by the authors is in black color

- A marked-up manuscript is added. Changed section with regard to the comments by reviewer 2 are marked in orange. General changes made by the authors are marked in green.

**Specific comments "S"**

1. "*The article is too long. Please try to present the results in a less repetitive way. It is quite hard to compare the results of the different configurations since they are presented separately. A more compact description of the results, focusing on the differences between the different VGs configurations would be highly appreciated.*"

The structure of the result part of the paper was revised according to the comment of the reviewer. A comparison between the different configurations becomes easier in this new structure.

2. "*It is well known that RANS has strong limitations for the analysis of separated flows, which is precisely the topic of this work. A Detached Eddy Simulation should actually be better suited for this kind of application. Please explain your reasons for using RANS and discuss the reliability of the results in this regard. If possible, I would recommend repeating at least some of the simulations with DES and comparing the results. This would help to assess the reliability of the RANS simulations.*"

The authors fully agree with the reviewer on the weaknesses of RANS when it comes to the simulation of physics of separated flow. Many publications have already shown the limits of RANS and VGs when it comes to stall. We have added this in our work (see R2:S2-a (page 5, line 132)).

Nevertheless, we decided to use URANS because the state of the art in the field of DES with VGs is not advanced enough. For instance, it is unclear what influence, the VG vortices, which operate in the boundary layer, have on the shielded RANS boundary layer. In addition, the VGs in our case are modelled by the BAY model and thus source terms are introduced into the boundary layer region. The influence of this on the shielding function is affected

with even more uncertainty than fully resolved VGs and should be studied separately through extruded airfoil simulations at first. The authors believe that the scope of such a study would require at least one precursor paper. To the best of the authors' knowledge, only Mereu et al. (https://www.sciencedirect.com/science/article/abs/pii/S0360544219316597) have studies VGs with DES methods (where the VGs were fully-resolved and not BAY-modeled). But their focus was not the behaviour of the shielding. A promising method would probably be Wall-Modelled LES (WMLES). In combination with a VSTG method (synthetic turbulence generator), the VG vortices could be fully resolved in LES without the risk of losses in the resolved boundary layer turbulence through modelled stress depletion. However, such a method is very expensive in computational costs for an entire blade due to the high mesh resolution requirements.

This said, and in the aim of shortening the manuscript, the authors would prefer to postpone the DES work into the future, as soon as the state of the art has reached the discussed maturity. We added a short explanation and try to emphasise the focus for future work on this topic (see $\boxed{\text{R2:S2-b}}$ (page 5, line 133)).

3. "*A grid independency study is missing. I suggest repeating at least some of the simulations with a finer grid in order to see how that influences the results.*"

Regarding the size of the Setup with 114.2 million cells and the large number of time steps the required computational budget will be very high for a convergence study. Except the refinement mesh the blade mesh is identical to the one of Wenz et al. (2022) who already presented a convergence study. This reference is already included in the paper (see $\boxed{\text{R2:S3}}$ (page 8, line 218)). Regarding the resolution of the refinement mesh, we used the results from 2.5D studies out of Seel et al. 2021 and Seel et al. 2022.

4. "*The article describes in different places (e.g. line 353) how the radial flow leads to rotational augmentation, which in turn increases the lift. But what is the effect on the drag force? The fact that the tangential force is increased does not necessarily imply that the rotational augmentation does not have any influence on the drag.*"

We did not focus on drag because the uncertainty with the BAY model regarding drag prediction is relatively high due to the inviscid modelling approach.

5. "*The conclusions section is in my opinion still a bit weak: what are the practical implications of this work? How can industry and science benefit from the main findings of the manuscript? Is it possible to give recommendations about the use, installation and limitations of VGs?*"

The practical implication and the benefit for the community is that the tuning of VGs on extruded airfoil sections in wind tunnels is not sufficient because the 3D effects are not considered. The only way to optimise the VG-related geometrical parameters on the rotor blade is clearly numerical simulation because experiments on entire rotor blades are almost impossible. Those two ideas are already formulated in a compact manner in $\boxed{\text{R2:S5}}$ (page 26, line 526) and important for both industries and science.

The authors tried to give as many general recommendations as possible out of this specific turbine, particularly in the root section, with one specific "best-practice" VG setup. Nevertheless, the authors are not able to give recommendations about installation and limitation of VGs beyond the already written aspects.

6. "*The authors claim that the VGs can lead under certain conditions to a torque increase of 3.28%. I think that it makes no sense to give results with two decimal positions when the uncertainty of the simulations is obviously much larger. I suggest stating „a torque increase of approximately 3%.*"

We completely agree and adapted the sentence (see $\boxed{\textbf{R2:S6}}$ (page 21, line 405)).

7. "*Line 60: I do not really understand this sentence. Mass conservation should apply. I suggest some rewording for explaining the idea.*"

Himmelskamp was only talking about the mass flow of the boundary layer, not of the entire slice. The authors reformulated it (see $\boxed{\textbf{R2:S7}}$ (page 3, line 60)).

8. "*The description of the CFD model is not detailed enough. Which type of discretization is used for time and space? Which are the boundary conditions? Some information on the computational efficiency of the model would be appreciated. How long did the simulations run with how many cores?*"

The information about the discretization were added (see $\boxed{\textbf{R2:S8-a}}$ (page 6, line 173) and $\boxed{\textbf{R2:S8-b}}$ (page 6, line 174)). The boundary conditions are already described in $\boxed{\textbf{R2:S8-c}}$ (page 6, line 165) and $\boxed{\textbf{R2:S8-d}}$ (page 8, line 211). The computational efficiency of the BAY-model is very high. Only the search algorithm for the VG cells which has to be computed once at the beginning has a significant overhead depending on the number of kernel points. In the aim of shortening the manuscript, a study concerning this efficiency is not added. The simulation costs in CPUh and number of cores were added (see $\boxed{\textbf{R2:S8-e}}$ (page 12, line 274)).

9. "*The size of the VGs is given meters, which in my opinion is quite difficult to read. I recommend giving it in mm.*"

The authors adapted this in table 2.

10. "*Line 296: I can not see the root vortex composed of two-counter rotating vortices. I suggest marking it as clearly as possible in the figure.*"

The authors adapted the Figure 8 (page 15) in order to see the direction of rotation of both vortices.

11. "*Line 326: This is an interesting interpretation of the role of root vortex. Can you relate it to Fig. 8 or any other visualization?*"

The authors related the Figure 8 (page 15) to the visualization (see $\boxed{\textbf{R2:S11}}$ (page 22, line 435)).

[revised manuscript text omitted]

---

## Author Comment (AC3)

**Reply to comments by Reviewer 3**

Ferdinand Seel on behalf of the authors
IAG, University of Stuttgart

June 23, 2023

The authors would like to thank the reviewer for his/her efforts and valuable comments. They are very much appreciated and incorporated into the revised paper.

In the present document, the comments given by the third reviewer are addressed consecutively. The following formatting is chosen:

- The reviewer comments are marked in blue and italic.

- The reply by the authors is in black color

- A marked-up manuscript is added. Changed section with regard to the comments by reviewer 3 are marked in red. General changes made by the authors are marked in green.

**Specific comments "S"**

1. "*L39: The BAY model is well known in the community of VG modelling, but it would still help if an explanation/description was given before the acronym was used. The audience of this paper will hopefully be broader than VG experts. Addition: i see this is provided later in the article. Perhaps then remove the BAY reference here? Is it needed?*"

The publication of Troldborg et al. (2016) is very important for BAY modelling as it was the first application to a full turbine blade. For this reason, the authors would like to cite the reference in the introduction. In the sentence, the authors specified the source term approach, but they tried to make it more clear now (see $\boxed{\text{R3:S1}}$ (page 1, line 10)).

2. "*L64: a clear definition of what is separated flow would be needed, before you can use the term 'nearly separated'. Please rephrase?*"

The authors rephrased the sentence (see $\boxed{\text{R3:S2}}$ (page 3, line 64)).

3. "*L100: depends what you mean by efficient here. They will prevent the flow from separating even if they extend beyond the BL height. In fact, the will probably be more effective as the shed vortex will be stronger. Perhaps rephrase or define what is efficient here?*"

The authors meant by "efficient" the avoidance of separation without the creation of excessive VG drag. The sentence was rephrased (see $\boxed{\text{R3:S3}}$ (page 4, line 101)).

4. "*Eq.2: how do you compute this velocity term? Is it an average over the VG area or something else? please clarify.*"

Thank you for this interesting question. We compute a trilinear interpolation of all the velocity vectors of the surrounded 27 cells onto the exact position of the kernel point. An explanation was added (see $\boxed{\text{R3:S4}}$ (page 7, line 194)).

5. "*L205: diameter? please specify*"

The authors adapted the sentence to make it more clear (see $\boxed{\textbf{R1:T1-b}}$ (page 8, line 220)).

6. "*L248: In fact they do increase Cl even before stall (see, among other studies, Wind Energy. 2018;21:745–765, Figure 13).*"

The authors adapted the sentence to make it more clear ($\boxed{\textbf{R3:S6}}$ (page 11, line 268)).

7. "*L248: Please explain in more detail. Is it de-cambering or have I misunderstood something?*"

It is an error, of course we meant de-cambering ($\boxed{\textbf{R3:S7}}$ (page 11, line 268)).

8. "*Fig. 5: Should there be two graphs or two curves for each parameter in this graph? (You state earlier that 'the local $\delta_{99}$ distribution along the radial direction zrel = z/R is different for each case')*"

The wording of "average" was misleading. The $\delta_{99}(z_{rel})$-distribution is different for the two cases. In order to get a good fit of the VG height for both cases, the authors decided to use the average of both curves for the adaption of the VG height. We changed the description of Figure 5 (page 11) and the wording in the text (see $\boxed{\textbf{R1:S7}}$ (page 10, line 242)).

9. "*The legend of the colorbar for the middle figure is so close to the x-axis of the top image, that can be confusing. Please rearrange either the figures or the color bars to provide better clarity. Also please consider increasing the size of this type of figures to cover the page width.*"

The authors rearranged the legend in all the plots. The plots are not covering all the page width because of the "manuscript" layout of WES for the review process. In the double column layout of the final paper, they will have the width of one column. This allows comparisons with other plots on the same page within the new arrangement of the final version (one section on blade loads and Himmelskamp and one section on the boundary layer).

10. "*Fig 7: I might be getting this wrong, but by visually examining fig. 7 I understand the following: From the Fy timeseries, there are at least two separate (?) dominant frequencies here. A 'slow' one with an average period of 2.3 sec appearing at z>0.1 and a 'fast' one appearing at z<0.1 with an average period of 1.15 sec. Intriguingly the periods have a ratio of two. I have tried to visually show this with the red squares in the Fy plot. I have pasted the same red squares to the dw/dt plot. I can see the 'slow' frequency, but instead of the fast one, I think an even faster one appears, with (T =0.5sec). Obviously, this is not the way to analyse this and a PSD spectrum would provide more insight. Have the authors looked into this? Finally, as w is an integral quantity, did the authors examine the dependence of the frequencies/phenomena on the extent and on the chordwise location of the WN region?*"

Thank you for this detailed comment. The authors agree on the fact, that there are two frequencies and adapted it in the text (see $\boxed{\textbf{R3:S12-a}}$ (page 13, line 298), $\boxed{\textbf{R3:S12-b}}$ (page 13, line 307), $\boxed{\textbf{R3:S12-c}}$ (page 13, line 310), $\boxed{\textbf{R3:S12-d}}$ (page 14, line 320) and $\boxed{\textbf{R3:S12-e}}$ (page 14, line 336)).

We intentionally decided not to do spectral analysis because the time samples are short enough to visualise it in time and see the differences between each period. Furthermore, a spectral analysis would require a much longer signal to reach statistical convergence of the detached flow. Such a computation would lead to excessive computational costs. Finally, the analysis of the blade without VGs was not meant to be the main focus of this publication and would stretch this section too much and would increase the length of the paper. The authors would

like to keep the focus of this publication on the interaction with VGs. The effects on the clean blade are too broad to address them in more detail than in the latest version. This could be the subject of a new publication instead.

Nevertheless, in order to give a deeper insight to the reviewer, the authors undertook the spectral analysis through a Fast-Fourier-Transform (FFT) of the DesNoVG case (see attached Figure **??**). The two discussed phenomena (i.e. slower and faster) are clearly visible through the FFT.

[Figure]

Figure 1: FFT of $F_y$ for the DesNoVG case over the radial direction.

Concerning the extent of the WN region: The region is selected much larger in spanwise direction than the VG arrays are, and thus covering the entire area of interest. Studies on the extent of the slice in spanwise direction were not undertaken. Regarding the chordwise location the strength of the phenomena increases for increasing chordwise position, but beyond 80% the analysis of the boundary layer (BL) was difficult as the BL edge could not be determined for all the spacial positions and timesteps.

11. "*L275: As the height of 3D separated flow is finite, the direction of the local velocity component is expected to vary significantly with distance from the blade surface. A: Was this considered? B: How did you select the final height for the WN70 zone? C: Velocity direction will also vary along the radial direction (or the length of WN70 slice). Is it possible that the large extent of the zone and the integral form of the metric 'masks' the unsteadiness? Was this considered?*"

A: It is true that the local velocity components are varying strongly with the distance from the blade surface. Hence, the absolute value of w is a mix between the relatively constant outer flow and the varying flow in the BL. This is one of the reasons why the derivative of w was used: The outer flow is relatively constant, and thus the unsteady flow phenomena of interest are well visible.

B: In wall-normal direction, the extent was chosen in order to include the highest observed separated structures for the case with the strongest flow separation (oDesNoVG).

C: The direction of velocity is varying, but our focus was on the spanwise velocity w. This component is rather low in the outer blade area. Furthermore, we did not observe any other unsteady phenomena which could mask the regarded detached flow (no inflow turbulence, no

tilt angle, no tower). Consequently, we do not expect any other unsteady phenomena than flow separation to mask the derivative in time of w.

12. *"L276: I think the discussion here can be improved by distinguish between faster and slower phenomena, see comment on fig. 7."*

The authors adapted the manuscript in the proposed way (see R3:S12-a (page 13, line 298), R3:S12-b (page 13, line 307), R3:S12-c (page 13, line 310), R3:S12-d (page 14, line 320) and R3:S12-e (page 14, line 336)).

13. *"L290: Here a phenomenon that has a frequency of 0.5 Hz is explained via snapshots only 0.4sec apart (f=2.5Hz). Please reconsider the discussion here. Also, consider the use of spectra in the analysis. Other than that the 0.4sec cycle is very nicely explained. "*

The snapshots aim to represent crucial stages of the phenomenon in more detail. The duration until a repetition of the entire phenomenon is around 2s and thus 0.5Hz.

The answer regarding the spectral analysis is discussed in detail in the reviewer comment number 10.

14. *"L298: This location is unclear in Fig. 8, please indicate"*

This is an error from our side. We meant Figure 7 (page 13). It was corrected in the manuscript (see R3:S14 (page 14, line 327))

15. *"L298: Which time intervals? This is unclear. Also the 'slow' period appears to be 4 times the 'fast' one, not three. Unless I have misunderstood something, in which case a rephrasing would help."*

The sentence was adapted (see R3:S15 (page 14, line 327)).

16. *"Could you please show the difference between the design and off-design cases with a Cp-x graph at z=0.09? I understand the analysis is 3D, but it would help visualise the extend of the 3D effects."*

Unfortunately, we had to limit the output during the computation and do only have some surface data files available. For this reason, an averaging of the surface data is not possible and a comparison of cp-x snapshots might be more misleading than clarifying.

17. *"Fig 8: 5.4, instead of 5,4, same for the figure below. Is the contour of the vorticity magnitude the optimal option here? The lack of direction can be misleading (see the vortices sketched here). What about streamwise vorticity (streamwise for the blade/rotating frame of reference, not the global)? "*

The figure was adapted (see Figure 8 (page 15)). The direction of the vortices was added. The streamwise vorticity would only show the root vortices, but the separated area has rather a spanwise vorticity, which should also stay visible in this plot. The authors added two curved arrows in order to represent the counterrotating root vortices.

18. *"Fig. 9: Again, I might be wrong, but it seems that the 'slow' phenomenon of T =2.3 sec is still present, whereas the faster one has disappeared or is less pronounced? I think a spectral analysis is would help a lot."*

For the reasons already discussed in reviewer comment 10 the authors would kindly ask the reviewer to not investigate deeper the noVG case. This is also in agreement with the wishes of the two other reviewers, who proposed to rather shorten the publication than do further investigations. Nevertheless, we agree totally that a spectral analysis would give more insight and further discussion.

For this case, the time sample is even shorter than the DesnoVG case by a factor of 2. Consequently, the statistical convergence is even worse. Nevertheless, the FFT is represented in Figure 2. As mentioned by the reviewer, the higher frequency phenomenon is clearly reduced, whereas the slower one is still pronounced.

[Figure]

Figure 2: FFT of $F_y$ for the oDesNoVG case over the radial direction.

19. *"I understand the explanation, but this contradicts present practice experience and existing literature. Currently a number of companies offer VG retrofitting services and OEMs design their new blades with VGs. Also, the literature already cited herein claims to have gains from VG application. (see e.g. http://arc.aiaa.org | DOI: 10.2514/6.2015-1035, Fig. 8) How do you explain the contradictory results? Is it a matter of VG design? Is it blade specific?"*

Thank you for this very interesting and relevant comment. As long as the blade is designed to be able to hold the flow completely attached (no soiling and/or no erosion) and the blade is operating at its design point, VGs do not provide considerable advantages, and they may even have a slight negative impact as they inhibit spanwise flow and create drag. Nevertheless, the authors would like to emphasise that this situation (no soiling, no erosion) is rather fictive because it is only relevant for a very short period of time at the very beginning of the lifetime of the wind turbine. Thus, VGs do provide advantages most of the time either as soon as they are included in the design (through higher pitch angles/relative thicknesses possible than for a blade without VGs and/or new airfoils designed to perform with VGs) or if the blade is not able any more to keep the flow attached. This is the case for erosion, which is counteracted by a retrofitting. Furthermore, VGs may also be advantageous in case of gusty wind conditions (e.g. complex terrain) as the relative angle of attack increases temporally which could lead to flow separation.

20. *"I think it would be useful to show how the other mode of the bimodal flow looks, here. So, a figure like Fig. 19 at t= 0.5sec?"*

The authors added the new figure (oDesVGin at t=0.5s see Figure 19 (page 24)) and included it into the text (see $\boxed{\text{R3:S20}}$ (page 21, line 412)).

[revised manuscript text omitted]